# Estimating the impact of interventions against COVID-19: From lockdown to vaccination

**James Thompson** [1] *, **Stephen Wattam** [2]

**1** Dept. of Mathematics, University of Luxembourg, Esch sur Alzette, Luxembourg, **2** W&P Academic Consultancy, Manchester, United Kingdom

* j.thompson@turing.ac.uk

## Abstract

Coronavirus disease 2019 (COVID-19) is an infectious disease of humans caused by severe acute respiratory syndrome coronavirus 2 (SARS-CoV-2). Since the first case was identified in China in December 2019 the disease has spread worldwide, leading to an ongoing pandemic. In this article, we present an agent-based model of COVID-19 in Luxembourg, and use it to estimate the impact, on cases and deaths, of interventions including testing, contact tracing, lockdown, curfew and vaccination. Our model is based on collation, with agents performing activities and moving between locations accordingly. The model is highly heterogeneous, featuring spatial clustering, over 2000 behavioural types and a 10 minute time resolution. The model is validated against COVID-19 clinical monitoring data collected in Luxembourg in 2020. Our model predicts far fewer cases and deaths than the equivalent equation-based SEIR model. In particular, with $R_0 = 2.45$, the SEIR model infects 87% of the resident population while our agent-based model infects only around 23% of the resident population. Our simulations suggest that testing and contract tracing reduce cases substantially, but are less effective at reducing deaths. Lockdowns are very effective although costly, while the impact of an 11pm-6am curfew is relatively small. When vaccinating against a future outbreak, our results suggest that herd immunity can be achieved at relatively low coverage, with substantial levels of protection achieved with only 30% of the population fully immune. When vaccinating in the midst of an outbreak, the challenge is more difficult. In this context, we investigate the impact of vaccine efficacy, capacity, hesitancy and strategy. We conclude that, short of a permanent lockdown, vaccination is by far the most effective way to suppress and ultimately control the spread of COVID-19.

## Introduction

The ongoing COVID-19 pandemic is among the most disruptive global events in modern history. At the time of writing, the SARS-CoV-2 virus has spread to almost every country in the world, resulting in over two hundred million infections and at least five million deaths. It is of vital importance that we continue to build a rigorous understanding of how the SARS-CoV-2 virus spreads and predict the impact of interventions, to help policy makers formulate effective strategies that save lives while simultaneously balancing the economic and social impact.

**Data Availability Statement:** All input and output data files are available in the public repositories https://github.com/abm-covid-lux/abmlux and https://github.com/abm-covid-lux/output. All input data can be found in the Scenarios/Luxembourg

folder of the abmlux repository while all output data can be found in the output repository.

**Funding:** This project was funded by a grant from the COVID-19 Fast-Track program of the Fonds National de la Recherche Luxembourg (https://www.fnr.lu/), with the grant being awarded to Dr. J. Thompson. The reference number for this project is: COVID-19/2020-2/14858807/ABMLUX The funder provided support in the form of salary for Dr. S. Wattam, but did not have any role in the study design, data collection and analysis, decision to publish, or preparation of the manuscript. The specific roles of authors are articulated in the 'Author Contributions' section. One of the authors, Dr. S. Wattam, is affiliated with a commercial company, W&P Academic Consultancy, but we declare that no competing interests resulted from this. In particular, this affiliation does not alter our adherence to PLOS ONE policies on sharing data and materials.

**Competing interests:** The authors have declared that no competing interests exist.

Central to such a strategy is a recognition of heterogeneity and behavioural diversity. Indeed, the regional impact of COVID-19 has been extremely variable. For a given region, the impact of any infectious disease of humans depends fundamentally on who lives in that region, how these individuals interact with one another and how this population connects with the populations of other regions. For a given individual, factors such as age, sex, ethnicity and the presence of underlying medical conditions might determine how that individual responds to an infection. Transmission is not only determined by the nature of the disease itself, but also by a multitude of factors relating to human behaviour. Such factors might include the time of day, the day of the week, the climate, seasonal effects and the prevailing culture of the region. These underlying variables result in correlations, producing an extremely complex and computationally irreducible system of social interactions and disease dynamics, beyond the scope of simple mathematical theory. Modelling the impact of public health policy, in the context of an infectious disease such as COVID-19, is therefore necessarily difficult and subject to unavoidable limitations.

One commonly used indicator of epidemic dynamics is the effective reproductive number $R_t$, defined roughly as the expected number of secondary infections caused by a typical infected individual at time $t$. This number aggregates the factors mentioned above by simultaneously averaging over individuals and individual behaviour. However, the usefulness of this quantity is subject to certain limitations [1] and, since it is defined in terms of a universal average, it is not possible to measure the true $R_t$ of a population. Additional simplifying assumptions, on the population and its mixing habits, are required in order that $R_t$ be estimated. The most basic assumption supposes that all individuals are identical and mix with one another with equal probability. In a sufficiently large population with sufficiently many individuals infected, such mass action might be realistic, but in circumstances where the proportion of infected individuals is low, it neglects the unpredictable nature of interactions between small numbers of people. Nevertheless, such homogeneity assumptions give rise to a number of popular mathematical and computational models, including the equation-based compartmental models [2]. Such models typically use ordinary differential equations to keep track of how many individuals are in various health states at various times, sometimes stratified by age or households. The $R_t$ associated to such a model can be fairly easily calculated, as well as certain other quantities of interest, for example limiting equilibria.

The equation-based approach to epidemic modelling could be considered the *top-down* approach, which postulates a set of equations whose solution, after appropriate configuration, is supposed to describe the system in question. Such an approach has the advantages of flexibility and speed, typically involving only a small number of parameters, but on the other hand is unable to capture the heterogeneity and granularity obtained using the *bottom-up* approach of an agent-based model. In an agent-based model, the simultaneous actions and interactions of multiple individuals, referred to as agents, are simulated in an attempt to re-create and predict the emergence of complex phenomena as a result of their collective behaviour.

Agent-based models are computationally intensive, and therefore have risen to prominence only in recent decades, with one of the earliest examples being John Conway's Game of Life [3]. Agent-based models have been applied across many areas of study, for example ecology [4], social science [5], macroeconomics and financial markets [6] and epidemiology. Agent-based models have been used extensively to study the spread of infectious diseases including COVID-19, as will be discussed in the next section.

In this article, we present an agent-based epidemiological model based on collocation. At each moment in time, our model partitions the population into subsets, with each subset corresponding to a particular location, for example a house, restaurant or shop. These subsets describe who is in each location at each time, with homogeneous mixing occurring internally.

As individuals move between locations, the subsets are updated accordingly. On top of this sits our disease model and a range of interventions. The model is custom-built, featuring numerous heterogeneous dimensions and behavioural diversity. It is able to capture both spatial and temporal variations in disease dynamics. The model consists of four basic layers, described as follows:

- **Locations**: A procedurally generated random environment of locations.

- **Agents**: A heterogeneous population with daily and weekly routines defined on a 10 minute time resolution.

- **Disease model**: An age-dependent compartmental model featuring hospitalization and intensive care.

- **Interventions**: Implementations of a broad range of public health interventions.

Interventions are the means by which a policy maker can control or suppress an epidemic. Interventions are either pharmaceutical or non-pharmaceutical. The World Health Organization divides the latter into four categories [7]. First there are the personal protective measures, which includes improved hand hygiene, respiratory etiquette and face masks. Second are the environmental interventions of improved ventilation and surface and object cleaning. Third are the various physical distancing measures, including such things as quarantining, school closures, workplace measures, closure of businesses, cancelling of events, curfews and lockdowns. Fourth are the travel-related measures, referring to travel advisories, entry and exit screening, internal travel restrictions and border closures. Various combinations of these interventions have been implemented by governments around the world in response to the COVID-19 pandemic, with testing and contact tracing systems being used to gather information on who is, or who might be, infected. Accompanying the non-pharmaceutical inventions are the pharmaceutical inventions, in particular anti-viral therapies and, perhaps most importantly, vaccination.

Vaccination is generally considered the most effective method of preventing infectious diseases, with mass vaccination campaigns having achieved the global eradication of smallpox and the suppression of diseases such as polio, measles and tetanus from much of the world, thereby saving hundreds of millions of lives. Controlling COVID-19 on a global scale cannot be achieved using only the non-pharmaceutical interventions listed above, associated to which are enormous economic and social costs, and therefore mass vaccination against COVID-19 must form a central part of any successful COVID-19 control strategy. There are well known mathematical models of the relationship between vaccination and herd immunity, for example [8].

Several COVID-19 vaccines have been developed and tested (see, for example, [9–12]) and are being distributed around the world. In most countries, vaccines are administered according to a priority list, starting with either those individuals who most require immediate protection against the disease, or those individuals for whom reduced transmission will be of the greatest benefit from a public health perspective. Besides the manufacturing and logistical challenges associated with mass vaccination, there is also the issue of vaccine hesitancy [13–15], which refers to the fact that significant numbers of people would prefer, for various reasons, not to get vaccinated. Assessing the impact of vaccination, against the backdrop of various overlapping non-pharmaceutical interventions, is therefore challenging.

The objective of this article is to compare interventions according to their epidemiological impact in the model. We consider, in particular, the following questions:

- How do non-pharmaceutical interventions compare, in terms of their impact on cases and deaths?

- At what level is herd immunity achieved?

- To what extent does the success of a vaccination campaign depend on factors such as efficacy, daily capacity and hesitancy?

- How does a vaccination strategy that focusses on reducing deaths compare to one that focusses on reducing transmission?

In this article, we are not concerned with the economic or social costs of the interventions. Moreover we do not look for optimal strategies, this being instead a topic for future research. Validation will focus on cases, hospitalizations and deaths, avoiding such things as the basic or effective reproduction numbers. We will measure the impact of interventions by comparing cases and deaths with those of the baseline scenario in which no interventions are active. We will suppose vaccination is implemented in a two-dose format, with an interval of time between doses and limited daily availability. Our interpretation of efficacy is that, with a certain probability, a dose delivers immunity that protects the individual against infection. We suppose there exists a priority scheme that administers doses to certain individuals before others, based on their age, living arrangements or place of work, and who potentially refuse the vaccine with a certain age-dependent probability.

Our model is configured to represent Luxembourg, a small western European country with a population on 1st January 2020 estimated at 626,108, together with populations of cross-border workers in the neighbouring countries of Belgium, France and Germany. Input data therefore comes from various institutions and surveys associated with Luxembourg. Consequently, this article investigates the impact of interventions specifically in Luxembourg, although the model itself is flexible and can be adapted to other regions. Luxembourg, however, is particularly interesting because, while being an independent nation with its own unique response to the COVID-19 pandemic, has a population small enough to be within the reach of a computational agent-based model.

All input data, code and all output data generated by the code, used to plot figures or otherwise underlying the results presented in this article, can be found in publicly accessible repositories in GitHub. Input data and code can be found at https://github.com/abm-covid-lux/abmlux while output data can be found at https://github.com/abm-covid-lux/output.

Census data, including data on age distribution and household structure, were obtained from STATEC, the government statistics service of Luxembourg [16]. Public transport data came from Mobilitéit [17] and the Ministry of Mobility and Public Transport (MMTP) of the government of Luxembourg [18]. Population grid data came from the 2011 GEOSTAT study, organized by Eurostat [19]. Location counts came from STATEC and OpenStreetMap [20]. Behavioural and mobility data came from the 2014 Luxembourg Time Use Survey [21] and the 2017 Luxmobil Survey [22], conducted by STATEC and MMTP, respectively. COVID-19 clinical monitoring data came from IGSS, the General Inspectorate of Social Security of Luxembourg [23]. Interventions were otherwise parametrized using public knowledge, for example on the timing of lockdowns.

The key unknown parameters in our model are the transmission probability, initial exposure count and asymptomatic probability. The calibration of these and other parameters is discussed in the model evaluation section.

The organization of the paper is as follows. In the next section, we briefly describe the state of the art, referencing a sample of articles from the immense body of work that has emerged since the start of the COVID-19 pandemic. In the section after we describe our model. This is

followed by a section on model evaluation, in which we discuss the processes of verification and validation and the limitations of the model. After that we present and discuss our main results. Finally, in the last section, we draw conclusions, while making further remarks about the limitations of the study and directions for future research.

## State of the art

Since the early days of the COVID-19 pandemic, many thousands of articles have been written about the SARS-CoV-2 virus and its spread throughout the world. While some have used models based on systems of differential equations to describe the spread of the virus, others have taken the approach of agent-based modelling, with some studies incorporating elements of both. Early models focussed on predicting the severity of the pandemic, the impact of social distancing restrictions and face masks and the consequences of lifting restrictions too early. As the first wave of cases passed, the focus then shifted towards the lifting of restrictions and the reopening of schools and universities. Later in 2020, the emergence of second waves and the onset of mass vaccination against COVID-19 led many to investigate the impact of vaccination specifically, in the context of rising cases and against the backdrop of other non-pharmaceutical interventions.

Among the great range of publications and preprints focussing on the impact of non-pharmaceutical interventions, consider [24–31] for examples of models based on systems of differential equations, and [32–53] for examples of agent-based models. A number of the agent-based models are open-source, including Covasim [54], OpenABM-Covid19 [55], COMOKIT [56] and JUNE [57]. Covasim and OpenABM-Covid19 assume individuals mix homogeneously outside households, workplaces or schools. On the other hand, COMOKIT and JUNE are somewhat more similar to our own model, using dynamic contact structures developed via mobility and daily agendas.

In addition to our own agent-based model of COVID-19 in Luxembourg, Laurent Mombaerts and Atte Aalto have also developed such a model, using social security data to construct a contact network. Their model has been used in the recently published article [58] to study the large-scale COVID-19 testing programme in Luxembourg. Epidemic models based on contact networks are popular since various mathematical tools from graph theory can be applied to such models, resulting in a topological or geometric analysis of the underlying network [59]. For other articles examining Luxembourg specifically, consider [60], in which the authors used an equation-based model to search for optimal strategies for lifting restrictions in Luxembourg and several other countries, using genetic algorithms and techniques from machine learning, or [61], in which the authors studied the interplay between the epidemiological and economic aspects of the COVID-19 pandemic in Luxembourg. In [62], the authors used a compartmental model to study the impact of interventions in Luxembourg, including vaccination.

With several vaccines having being developed against COVID-19, a large number of articles have been written investigating their potential impact. Consider, for example [63–74]. In [65, 72], the impact of vaccination on cases, hospitalisations and deaths was studied using agent-based models, these two articles focussing on areas in the United States and Canada, respectively. The authors of both articles assumed a predetermined coverage rate achieved by the vaccination campaign and a specific vaccination rate of 30 individuals per 10,000 population per day, with efficacy against symptomatic infection set to 95%. Various levels of pre-existing immunity were also assumed, ranging from 5% to 20%, depending on the region. In [75], the authors studied the optimal choice of vaccination strategy under a partial or complete lockdown. Each of the individuals appearing in their model had a pre-assigned daily routine,

specified on the resolution of 1 hour, with the routine determining the order in which the individuals move between different locations, such as workplaces, schools, public places, hospitals and homes. The effect of vaccination combined with non-pharmaceutical interventions was also studied in [66], for the state of North Carolina, for interventions including reduced mobility, school closure and face mask usage. That article investigates scenarios under which vaccine efficacy takes the values of 50% or 90%. In our vaccination simulations, we will also consider variable efficacy, pre-existing immunity and limited daily capacity, together with other factors such as vaccine hesitancy.

An extension of the present work would be to capture the economic and social costs of interventions and apply tools from machine learning to search for optimal strategies, as the authors of [60] did for their equation-based model. Machine learning has found a wide array of applications in the context of the coronavirus pandemic, both when building models from training data and feature extraction but also when searching for optimal strategies. The article [76] provides a review of deep learning applications for COVID-19, in the context of not only epidemiology but also natural language processing, computer vision and life sciences. For other applications of deep learning to the COVID-19 pandemic, see [77, 78]. In many studies, it is assumed that all individuals comply perfectly with non-pharmaceutical interventions, while in reality this may not be the case. Adaptive agent behaviour can be accomplished using techniques from machine learning. See also [79] for a study of the effects of social learning on the transmission of COVID-19 in a network model. An approach utilising Bayesian techniques and a game theoretical modelling of adherence to restrictions was applied in [80], while the use of game theory and social network models for decision making on vaccination programmes was further emphasised in [81]. For simplicity, we will assume perfect compliance with interventions, except face masks and vaccination.

Some models have been formulated in terms of stochastic differential equations. For example, the article [82] presented an approach to modelling spatio-temporal vaccination strategies, wherein individuals move within a continuous space according to Brownian motion dynamics and, when they find themselves within a certain distance of one another, interact and potentially transmit the virus. Our model also features spatial dimensions, and could therefore also be used to investigate spatial strategies, for example ring vaccination, however this is beyond the scope of the present study.

In comparison to other epidemiological models of COVID-19, ours is more detailed and dynamic than most, containing a wide range of locations, a fine time resolution of only 10 minutes and over 2000 behavioural types. These features allow our model to track the experiences of individual agents and capture the sort of brief encounters that take place outside of homes, work and schools, for example in shops and restaurants. A simpler model would fail to capture an equivalent level of heterogeneity, and would not allow for such intuitive implementations of interventions. For example, a lockdown is achieved in our model simply by sending agents home, rather than by estimating a reduction in daily contact counts or the effective reproduction number. Contact counts are a not an input of our model, but rather an output. Our model contains a broad set of interventions, including vaccination, and is the first agent-based model to be applied directly to the study of mass vaccination against COVID-19 in Luxembourg.

## Methods

Our model is written in Python. The code is organized around a modular framework, in which components represent submodels. This has the advantage that new components, such as additional interventions, can easily be added while existing components can be quickly

updated or replaced. A communications system handles messages sent between the various components, a crucial feature since many of the interventions are required to interact with one another, while a scheduling system handles the timing of events such as lockdowns and testing regimes. The code is open source and available on GitHub at https://github.com/abm-covid-lux/abmlux.

All input data is found in a single configuration file separate from the rest of the code. The file, config.yaml, can be found in the Scenarios/Luxembourg folder of the abmlux repository, and indicates precisely which values are taken by each of the parameters appearing in the model. Using this file we are able to configure the model to represent different scenarios. The configuration files corresponding to the scenarios appearing in the model evaluation and results sections can be found in the output repository https://github.com/abm-covid-lux/output. The model is flexible, but as with most agent-based models [83] has the limitation of long run times for large populations.

We will now present an overview of the various layers of the model. A description of the model according to the ODD protocol [84] can be found in the appendix.

## Location types

The lowest layer of the model consists of a procedurally generated random environment, consisting of locations categorized by type. The list of location types includes:

- **Houses, Care Homes, Hotels, Primary Schools, Secondary Schools, Restaurants, Shops, Hospitals, Medical Clinics, Places of Worship, Indoor Sport Centres, Cinemas or Theatres, Museums or Zoos, Cars, Public Transport, Outdoors**.

  The remainder of the list consists of other types of working location, categorized by sector:

- **Agriculture, Extraction, Manufacturing, Energy, Water, Construction, Trade, Transport, Catering and Accommodation, ICT, Finance, Real Estate, Technical, Administration, Education, Entertainment, Other Services**.

To model Luxembourg, location counts are derived from a number of different sources. Table 1 lists the location counts for types for which we use data from OpenStreetMap (OSM), a collaborative project that aims to build a free editable map of the world.

The numbers of primary and secondary schools, as well as other working locations categorized according to sector, are estimated using data from STATEC, the government statistics service of Luxembourg. These numbers were published in the 2019 edition of their Répertoire

**Table 1. Estimated location counts in Luxembourg derived from OSM data, 2020.**

| Location type | Count |
|---|---:|
| Care Home | 52 |
| Restaurant | 2247 |
| Shop | 3136 |
| Medical Clinic | 125 |
| Hospital | 11 |
| Hotel | 213 |
| Place of Worship | 677 |
| Indoor Sport | 199 |
| Cinema or Theatre | 34 |
| Museum or Zoo | 77 |

**Table 2. Estimated location counts in Luxembourg derived from STATEC data, 2020.**

| Location type | Count |
|---|---:|
| **Primary School** | 181 |
| **Secondary School** | 58 |
| **Agriculture** | 86 |
| **Extraction** | 11 |
| **Manufacturing** | 785 |
| **Energy** | 98 |
| **Water** | 71 |
| **Construction** | 4366 |
| **Trade** | 4684 |
| **Transport** | 1349 |
| **Catering and Accommodation** | 251 |
| **ICT** | 2752 |
| **Finance** | 1241 |
| **Real Estate** | 1458 |
| **Technical** | 8349 |
| **Administration** | 2433 |
| **Education** | 838 |
| **Entertainment** | 87 |
| **Other Services** | 1523 |

des Entreprises Luxembourgoises [85]. Some care was taken to avoid overlap with working location types already listed above, the adjusted estimates being tabulated below in Table 2.

In addition, schools are divided into classrooms. In the case of Luxembourg, STATEC data indicates that, on average, each primary school consists of 17 classes while each secondary school consists of 34 classes.

Some locations types do not appear in these tables and are subject to special treatment. For example, the number of units of public transport is variable. A unit of public transport is defined to be either a bus or a carriage deck of a train or tram. A single-deck carriage consists of one unit, while a double-deck carriage consists of two units. The total number of buses and rail compartments operating in Luxembourg is derived from publicly accessible timetable data published by Mobilitéit. We used data referring to the period starting on 4th November 2019 and ending on 14th December 2019. Estimating average units per train at 10, average daily public transport availability in Luxembourg can then be visualized as in Fig 1. This determines the number of accessible locations of type **Public Transport**.

There is also a single outdoor location **Outdoor**, in which we assume zero disease transmission, and a **Cemetery**, to which agents are moved after death. In the Luxembourg implementation, there are also three border country locations, namely **Belgium**, **France** and **Germany**.

The number of locations of type **House** is determined by an algorithm that assigns agents to homes, to be described later. The number of locations of type **Car** is set equal to the number of houses, with each house being assigned one car. As with the units of public transport, the cars in our model are, for simplicity, static. The cars are simply locations in which agents are placed should they wish to use a car. In particular, agents living in the same house use the same car, no matter their destination. If an agent chooses to use public transport, then a unit of public transport is randomly selected among all those available at the time.

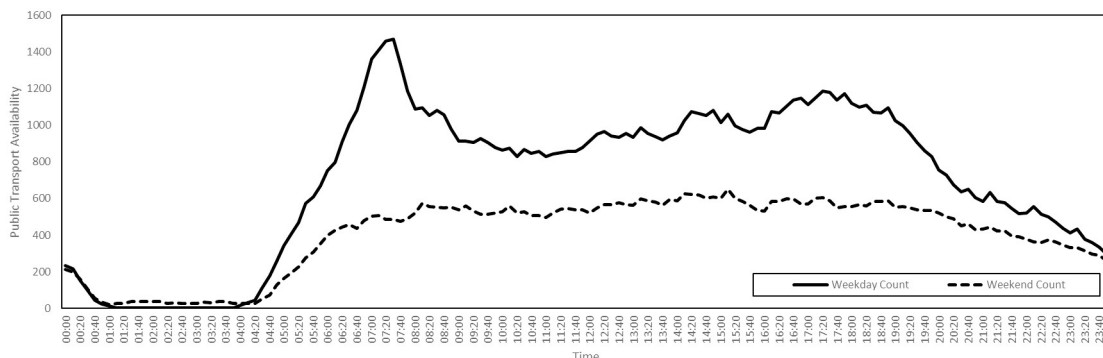

**Fig 1. Public transport availability on a typical day in Luxembourg, 4th November 2019 to 15th December 2019.**

## Spatial distribution

Locations are assigned spatial coordinates, as illustrated in Fig 2 against a map of Luxembourg and its 12 cantons [86]. By doing so, our model is able to capture the spatial dynamics of an epidemic.

To assign these coordinates, we first place a 1km square grid over the region in question, in our case Luxembourg. The grid we use is that of the ETRS89 reference frame. Each square is then assigned a weight, given by the number of people living in that square, according to Eurostat's 2011 GEOSTAT initiative. Then, for each location we select a grid square by sampling this weighted two-dimensional distribution. This specifies the coordinates of each location to a resolution 1km. To determine the coordinates to a resolution of 1m, we then sample uniformly within the 1km square. The coordinates are used by agents when choosing which locations to visit. We assume agents will tend to prefer locations close to, for example, their home.

We also have the option of sub-sampling the grid data to produce a grid of finer resolution. For example, with a resolution factor of 2, each original square with edge length 1km is replaced by four smaller squares each of edge length 500m. Population is then distributed among the small squares by linearly interpolating, with the option of setting the population of a small square equal to zero if there was no population present in the original square. Our population distribution model for Luxembourg, obtained using a resolution factor 2 and areas of zero population preserved, is illustrated as a heat map in Fig 2, together with a sample distribution of locations.

Since we set the spatial coordinates of a location by sampling the (interpolated) population distribution, we assume that all types of location are distributed as population is distributed. While this is approximately true, some location types are, in reality, subject to additional clustering. An improvement to the model would be therefore to assign coordinates using type specific spatial distributions, possibly achieved using additional OSM data, to produce a more realistic environment.

## Agents

Having generated a static environment of locations, we then populate this virtual world with agents. The agents in our model represent individuals. Agents are assigned a country of residence and an age. We do not assign sex, ethnicity nor the presence of underlying medical conditions.

Age is distributed according to the population of the region in question. In the Luxembourg model, age is distributed as in Fig 3, this data having been collected by STATEC, representing

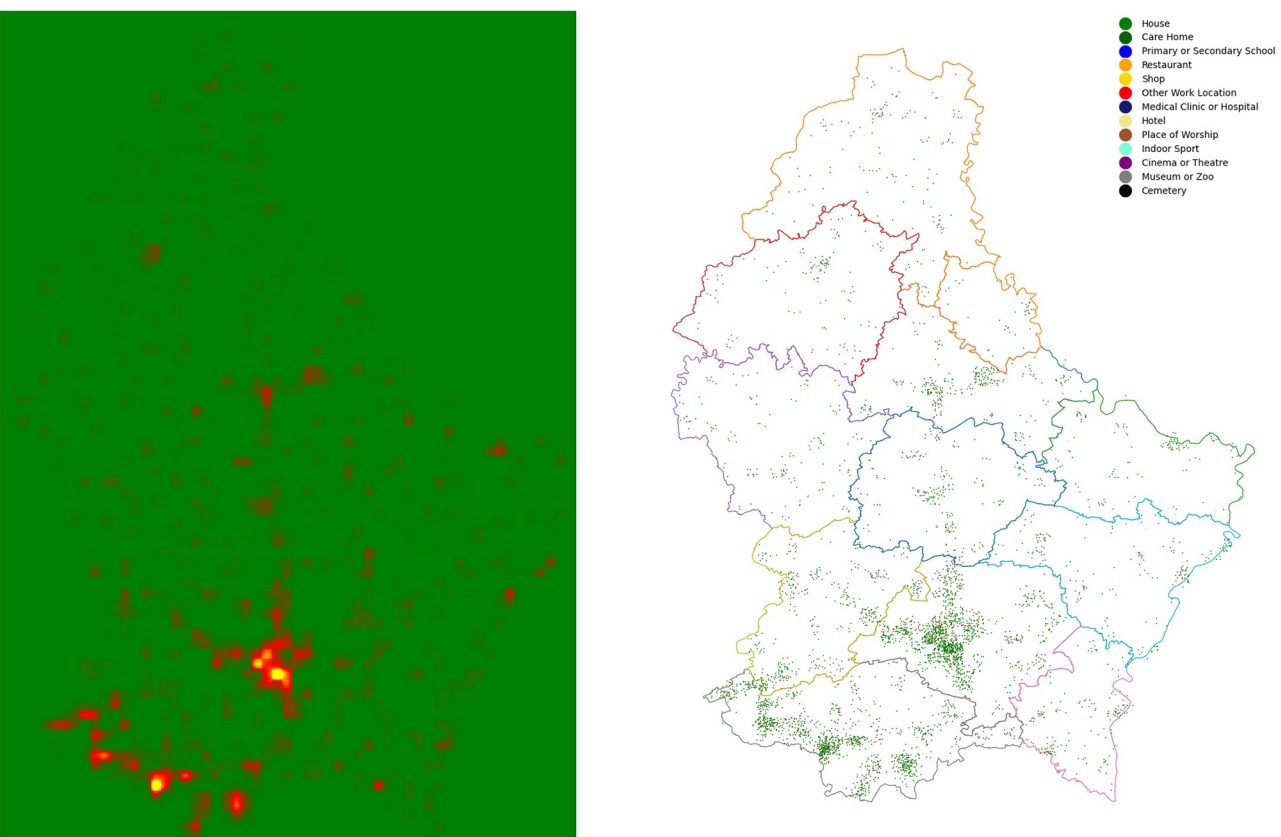

**Fig 2. Population distribution in Luxembourg, 2011.** On the left, a heat map of population in Luxembourg. On the right, 10,000 locations of the various types appearing in our model, distributed randomly according to the weighting illustrated on the left.

a resident population of 626,108 on 1st January 2020. We have suppressed the age category 95+ to 95.

In addition to the resident population, we also generate populations of non-resident commuters who live in neighbouring countries. Luxembourg shares borders with Belgium, France and Germany and large numbers of people travel across these borders every day for a variety of reasons. We focus on those who cross the border for work, since these are the individuals

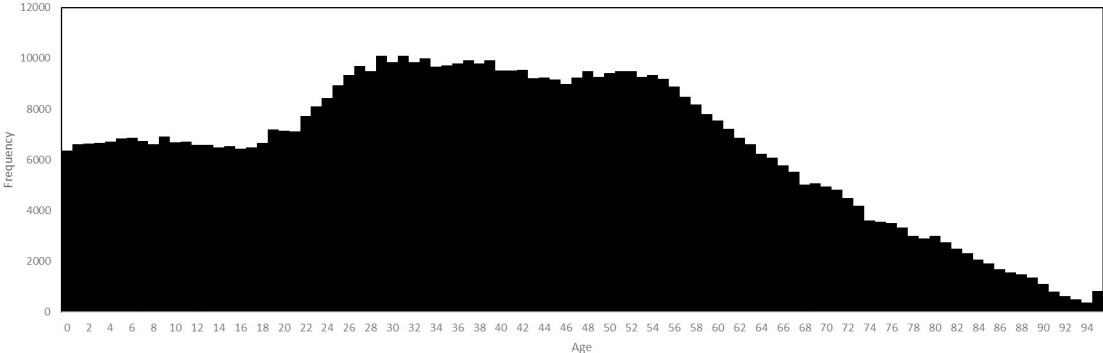

**Fig 3. Age distribution in Luxembourg, 1st January 2020.**

**Table 3. Populations of cross-border workers employed in Luxembourg, 2019.**

| Country | Workers |
|---|---|
| **Belgium** | 47173 |
| **France** | 104070 |
| **Germany** | 46863 |

who typically spend large amounts of time in the region and who travel on a regular basis. We assume that populations of cross-border workers consist only of adults, that the age of cross-border workers is distributed identically to that of adults in the resident population, and that cross-border workers travel to the region for work and for no other reason. We do not model air travel nor other long distance connections between regions. According to STATEC, the numbers of cross-border workers travelling to Luxembourg are given in Table 3.

## Activity choice

Agents are able to perform various activities. Activity selection is based on time use data. The Harmonised European Time Use Surveys (HETUS) [87] are national surveys conducted in European countries to quantify how much time people spend on various activities, including paid work, household chores and family care, personal care, voluntary work, social life, travel and leisure. Similar data are collected in other parts of the world, for example the United States. Respondents to the European surveys were asked to record dairies of both a week day and a weekend day, with a time resolution of 10 minutes. In other words, for each respondent, the time use data specifies what the respondent was doing during each 10 minute interval of each day. The list of activities recognised by the survey is long and therefore simplified for our purposes, resulting in the following list of activities appearing in our model:

- **Home, Visit, Work, School, Restaurant, Shopping, Outdoors, Car, Public Transport, Medical, Worship, Indoor Sport, Cinema or Theatre, Museum or Zoo**.

The activity **Home** refers to all domestic activities, such as cleaning, cooking and sleeping. The activity **Outdoors** includes such things as going for a walk, riding a bike or playing outdoor sports. The activity **Visit** refers to visits of family or friends in other houses or care homes. The activity **Medical** refers to medical activities not related to the epidemic, and places agents either in hospital or a medical clinic. The other activities are self explanatory. We construct weekly routines by concatenating 2 copies of the weekend dairy with 5 copies of the weekday diary for each respondent, with the week starting on a Sunday. We therefore do not distinguish between Saturday and Sunday nor between weekdays. In the Luxembourg implementation, data is derived from the 2014 Luxembourg Time Use Survey. The resulting distribution of activities performed each week is illustrated below in Fig 4. Differences between weekend and weekday behaviour are clear, as are features such as rush hour, lunch breaks and increased time spent outdoors at the weekend.

Since the age of respondents in the HETUS is known, we can assign agents weekly routines according to age. We do so by associating to each resident agent the routine of a respondent belonging to the same 10-year age bracket, randomly selected according to the statistical weights attached to data. This results, in the Luxembourg implementation, in over 2000 unique behavioural types. The minimum and maximum ages of respondents to the HETUS are 10 and 75, respectively, and we therefore introduce special rules for the very young and very old, in order to produce what we believe is a reasonable behavioural model covering agents of all ages.

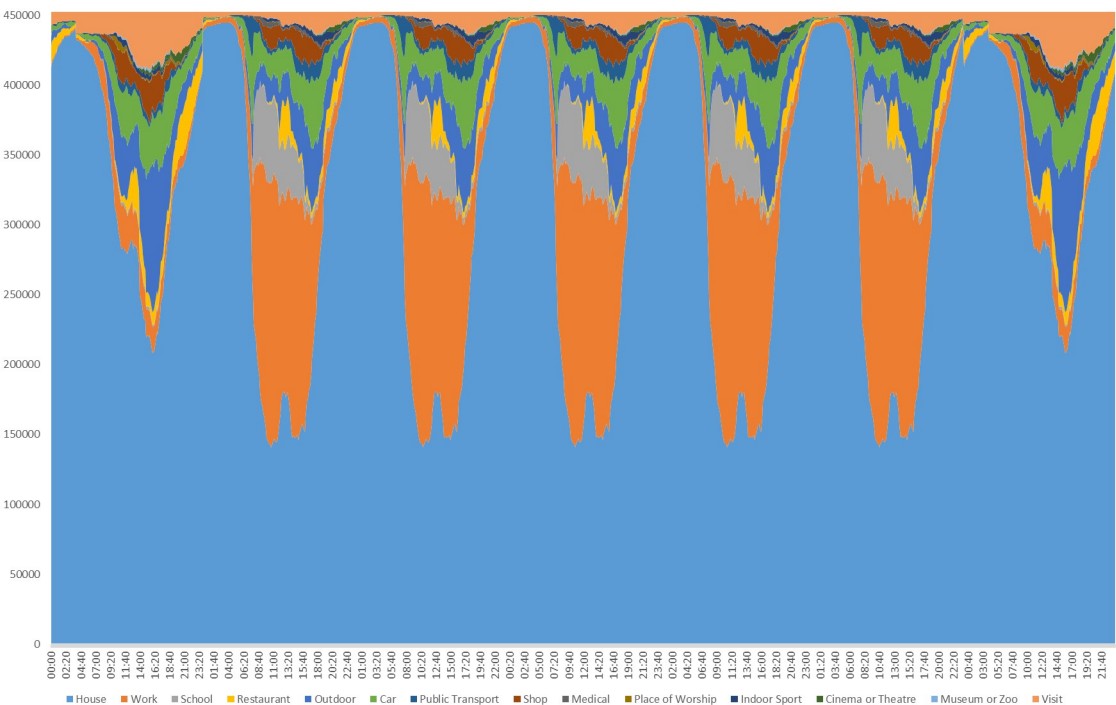

**Fig 4. Distribution of activities performed each week.** The week starts on a Sunday and the unit of the vertical axis is statistical weight.

Since the resolution of the time use data is 10 minutes, a weekly routine can be thought of as a vector of length 1080, with entries specifying which activity is to be performed at each corresponding time. For example:

$$[\textbf{Home}, \textbf{Home}, \textbf{Work}, \textbf{Work}, \cdots, \textbf{Restaurant}, \textbf{Home}].$$

Each agent is assigned such a vector. We can put a distance on the space of all such routines by summing the number of entries in which the activities of two routines differ. Doing so we can perform hierarchical clustering to determine if there exist naturally occurring behavioural types. A distance threshold of 250 yields a total of 358 clusters, the three largest of which, labelled 77, 147 and 176, are illustrated below in Fig 5.

Cross-border workers are assigned the canonical working routine given by the medoid of Cluster 77. This ensures that cross-border workers really do cross the border and go to work, since random sampling would have many of them performing other activities instead.

We also experimented with a more complicated activity model where agents choose activities randomly. This involved aggregating routines in such a way as to produce transition matrices and corresponding time-inhomogeneous Markov chains, the sampling of which generates infinitely many behavioural patterns. The drawbacks of this approach are the computational cost and the possibility of sampling unrealistic routines, so for simplicity we decided to stick with the deterministic system described above, in which agents read off which activity to perform directly from their given routine vector.

Having selected a preferred activity, an agent must then decide where to perform that activity. For example, if an agent decides to go **Shopping**, then the agent must choose a **Shop** at which to do the shopping. Agents are grouped into households and assigned a place of work, together with sets of locations at which they can perform the other activities.

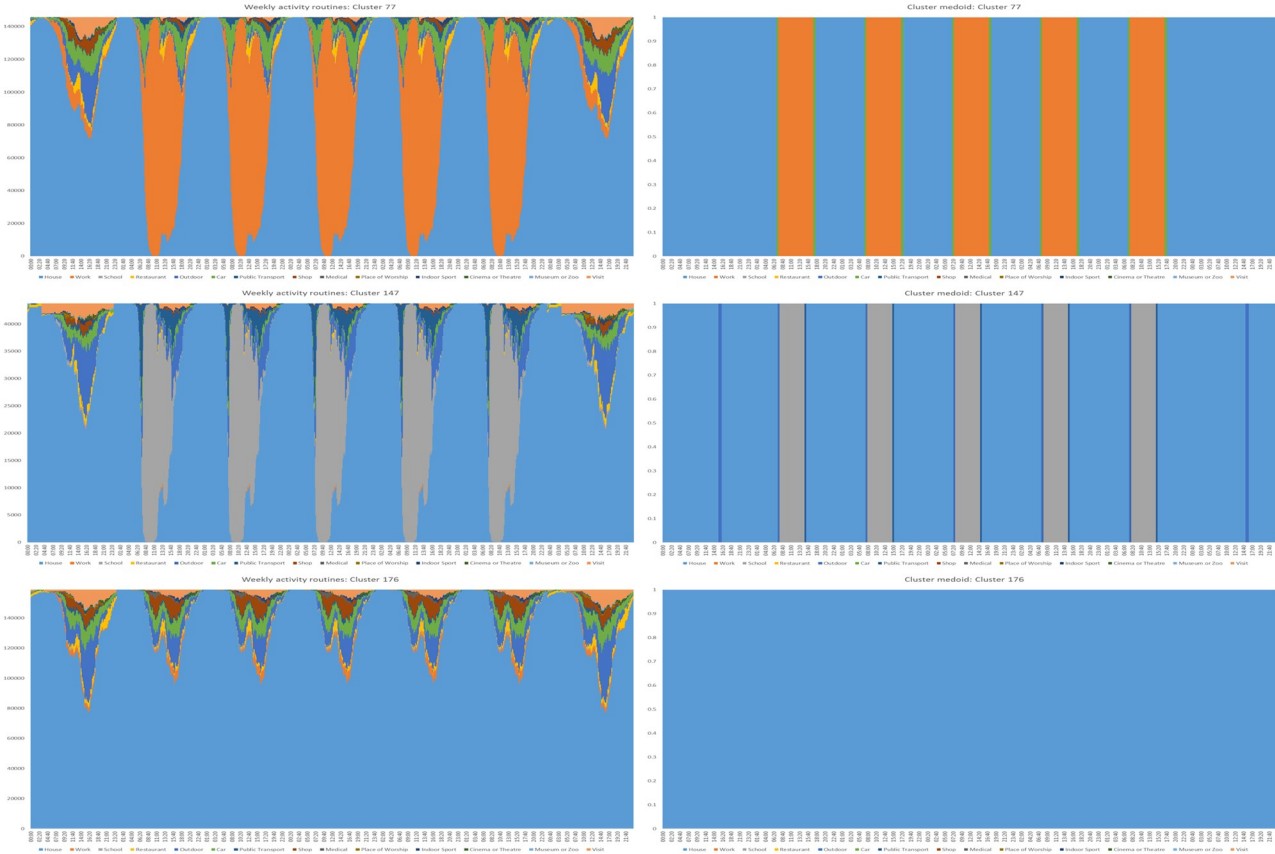

**Fig 5. Routine clusters.** On the left, the weekly routines of the three largest clusters with, on the right, the corresponding medoid routines.

## Households

The home of an agent is the location in which they perform the activity **Home**. Home assignment begins by populating care homes with the oldest residents and by setting the home of non-residents to be their country of origin. We assume that each care home contains 38 residents. We will assume that no internal transmission occurs within the neighbouring countries, focussing instead on transmission within the central region only. Remaining resident agents are then assembled into households, with household composition for the Luxembourg model being determined using population structure data on families and households collected by STATEC for the 2001 census. Data on the numbers of children and retired individuals in houses of various sizes in Luxembourg is tabulated below, in Table 4.

Note that in our implementation, the categories 5+ and 7+ are suppressed as 5 and 7, respectively. The largest private household in our model of Luxembourg is therefore of size 7. Using only the data contained in these tables, we are able to construct a discrete probability distribution on household types. For a household of size $n$, a household type is a triple $(c, a, r)$ where $c$, $a$ and $r$ denote the numbers of residents in the ages categories 0-14, 15-64 and 65+, respectively, with $c + a + r = n$. For example, a household of size 5 containing two children, two adults and one retired person would be encoded $(2, 2, 1)$. If $N$ denotes the total number of households in the census data, with $C_n(c)$ and $R_n(r)$ the numbers of households of size $n$ with $c$

**Table 4. Household structure in Luxembourg, 2001.**

| | | Size of Household | | | | | | |
|---|---|---|---|---|---|---|---|---|
| | | 1 | 2 | 3 | 4 | 5 | 6 | 7+ |
| Children aged 14 years or less | 0 | 50384 | 46191 | 15885 | 9035 | 2514 | 784 | 276 |
| | 1 | 0 | 2382 | 12257 | 4972 | 2041 | 514 | 149 |
| | 2 | 0 | 0 | 1108 | 14021 | 2180 | 720 | 205 |
| | 3 | 0 | 0 | 0 | 253 | 4165 | 622 | 219 |
| | 4 | 0 | 0 | 0 | 0 | 38 | 737 | 150 |
| | 5+ | 0 | 0 | 0 | 0 | 0 | 5 | 146 |
| Persons aged 65 years or more | 0 | 33586 | 30677 | 25001 | 26510 | 9743 | 2706 | 821 |
| | 1 | 16798 | 6893 | 2388 | 1240 | 971 | 445 | 191 |
| | 2 | 0 | 11003 | 1656 | 483 | 207 | 214 | 112 |
| | 3 | 0 | 0 | 206 | 36 | 11 | 12 | 19 |
| | 4 | 0 | 0 | 0 | 12 | 4 | 3 | 0 |
| | 5+ | 0 | 0 | 0 | 0 | 1 | 2 | 2 |

children and $r$ retired, respectively, then we postulate that

$$\mathbb{P}((c, a, r)) = \frac{C_n(c)R_n(r)}{2N} \left( \frac{1}{\sum_{i=0}^{n-r} C_n(i)} + \frac{1}{\sum_{j=0}^{n-c} R_n(j)} \right)$$

where $\mathbb{P}((c, a, r))$ denotes the probability of the profile $(c, a, r)$ occurring. Note that this does indeed yield a discrete measure with unit total mass. During the initialization phase of our model, houses are generated with profiles sampled from this distribution and populated with appropriate numbers of agents taken randomly from the three age groups. Houses are spatially distributed as the other locations, according to interpolated population grid data.

## Location choice

After home assignment, agents are then assigned a place of work, to which they move if performing the activity **Work**. First, for each agent $i$, a subset $S_i$ of all working locations is sampled uniformly at random. Working with only a subset reduces the computational cost of the next step, which involves assigning to each workplace $l$ in $S_i$ a weight, obtained by multiplying together two subweights. The first is given by the expected number of workers $m(l)$ at location $l$, configured for the Luxembourg model using STATEC data published in the 2019 version of their Répertoire des Entreprises Luxembourgoises. The second is determined using mobility data and the Euclidean distance $d(l, h_i)$ to the agent's home $h_i$. The mobility data comes from the 2017 Luxmobil Survey, in which respondents were asked to record how far they travelled (in terms of network distance) when doing so for various reasons. We have plotted aggregations of this data, for a selection of activities including **Work**, in Fig 6.

Converting network distance to Euclidean distance using a detour ratio formula [88], the distributions plotted in Fig 6 yield, for each activity indicated, a step function $\omega_{\text{act}}$, that assigns to each distance the probability of an agent travelling that far when performing the corresponding activity. Then, for the activity **Work**, we assume that the probability $\mathbb{P}_{i,\text{Work}}(l)$ that agent $i$ is assigned location $l \in S_i$ as their place of work is given by

$$\mathbb{P}_{i,\text{Work}}(l) = \frac{m(l)\,\omega_{\text{Work}}(d(l, h_i))}{\sum_{l \in S_i} m(l)\,\omega_{\text{Work}}(d(l, h_i))}.$$

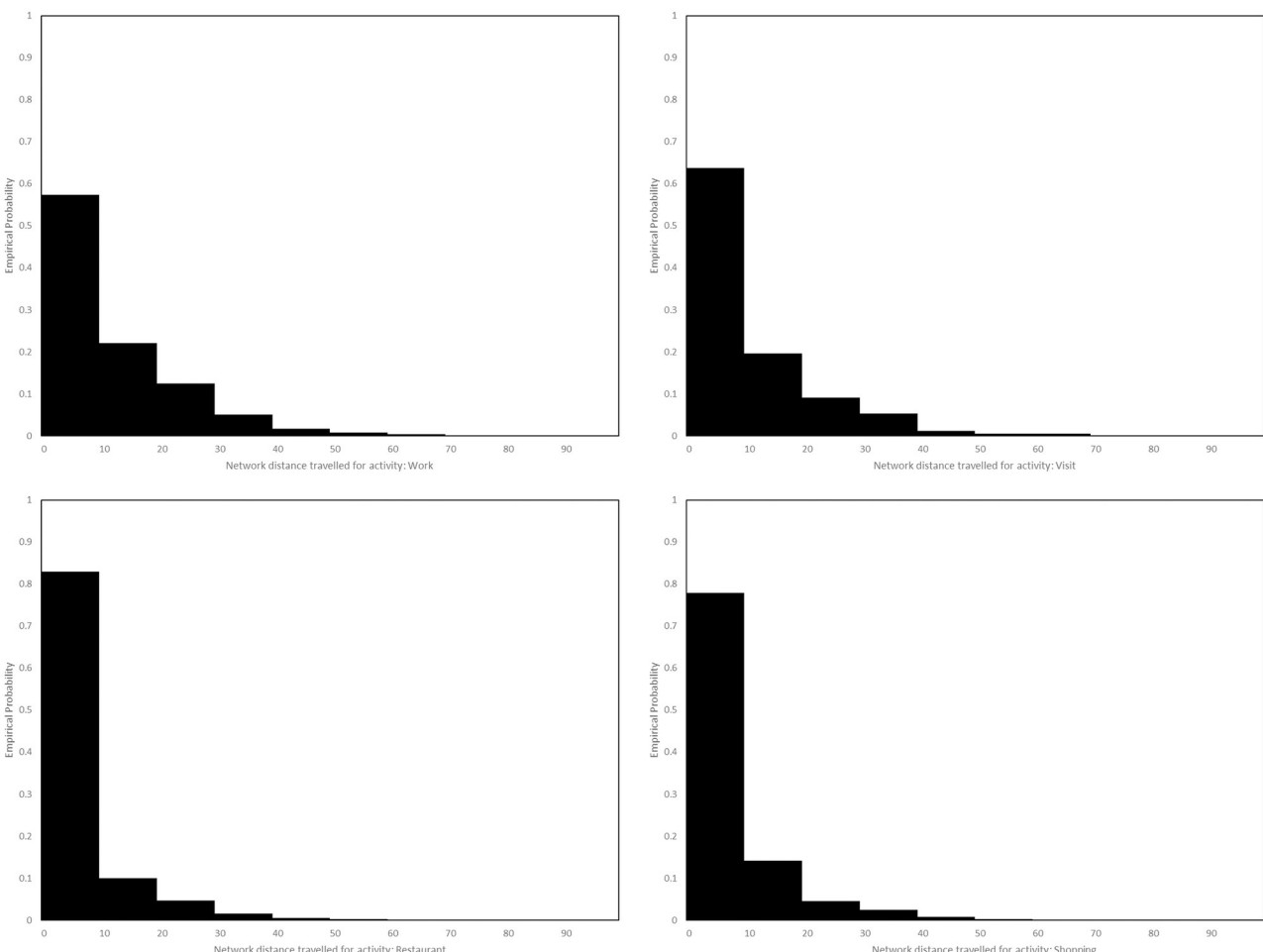

**Fig 6. Distributions of travel distance.** Distributions of network distance travelled in Luxembourg, in kilometres, for the activities **Work**, **Visit**, **Restaurant** and **Shopping**, according to the Luxmobil Survey, 2017.

For the activities **Shop**, **Restaurant** and **Visit**, we assume that the corresponding probability distributions $\mathbb{P}_{i,\text{act}}$ are defined similarly, but discarding the factor $m(l)$. Locations for some activities not specifically covered by the Luxmobil Survey, namely **Public Transport**, **Cinema or Theatre** and **Museum or Zoo**, are selected uniformly at random. Locations for activities **Schools**, **Medical**, **Worship** and **Indoor Sport**, are chosen based on household proximity. In the case of schools, we assume that if a school is full then the next nearest school is selected instead, ensuring that classroom sizes are uniform across the region. Moreover we assume that children from the same household attend the same primary and secondary schools.

Having completed this process, each agent then has, for each activity, a list of locations at which they are able to perform that activity. Preselecting locations in this way avoids the computational cost of distance-based calculation during the simulation. In Fig 7, we plot illustrations of these location assignments for three randomly selected agents. Where multiple locations exist for a given activity, the agent will choose randomly between them during the simulation. We assume that agents only move to a new location when starting a new activity.

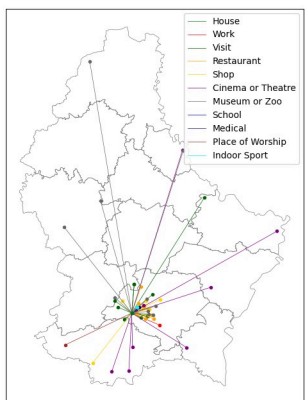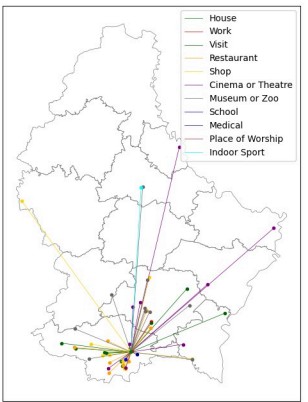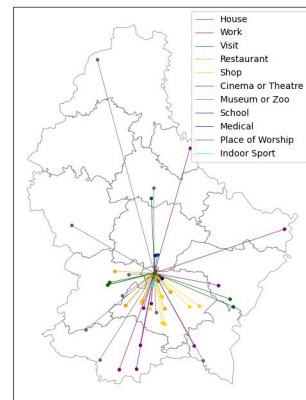

**Fig 7. Examples of agent networks.** For three randomly selected agents, we plot the locations at which the agents are able to perform the coloured activities, together with lines illustrating the distance to the agent's home. The agents are able to travel between these locations in any order. We omit the outdoors, since there is only one such location, cars, since an agent's car is placed at their house, and public transport, since the numbers are variable. Note that some locations appear in the networks of two or all three of these agents.

## Disease and transmission

Our disease model follows the SEIRD framework with additional compartments. The health states are characterized as follows:

- **Susceptible**: The agent is able to catch the virus.

- **Exposed**: The agent has caught the virus but is not yet infectious.

- **Asymptomatic**: The agent is infectious but not symptomatic.

- **Pre-clinically Infectious**: The agent is infectious but not yet symptomatic.

- **Clinically Infectious**: The agent is infectious and symptomatic.

- **Hospitalized**: The agent should be in hospital but not intensive care.

- **Intensive Care**: The agent should be in intensive care.

- **Recovered**: The agent has survived the disease and is no longer infectious.

- **Dead**: The agent has died of the disease and should be moved to the cemetery.

The model is visualized below in Fig 8, where arrows illustrate possible state transitions.

Using the first letter in the names of each health state, we encode the possible pathways through the above diagram as follows:

**SEAR**, **SEPCR**, **SEPCD**, **SEPCHR**, **SEPCHD**, **SEPCHIHR**, **SEPCHID**

For example, the pathway **SEPCD** describes an agent who having caught the virus passes through stages of pre-clinical and clinical infectiousness before dying from the disease outside of hospital. We assign to each agent a pathway, with probabilities determined by age. For the model of Luxembourg, these probabilities are derived from COVID-19 surveillance data managed by the General Inspectorate of Social Security in Luxembourg, collected during the first wave of COVID-19 cases in 2020. The age-dependent probabilities of an agent following the symptomatic pathways, conditional on that agent being symptomatic, are plotted in Fig 9. The probability that an agent follows the asymptomatic trajectory **SEAR** will be discussed later, in the subsection on model validation.

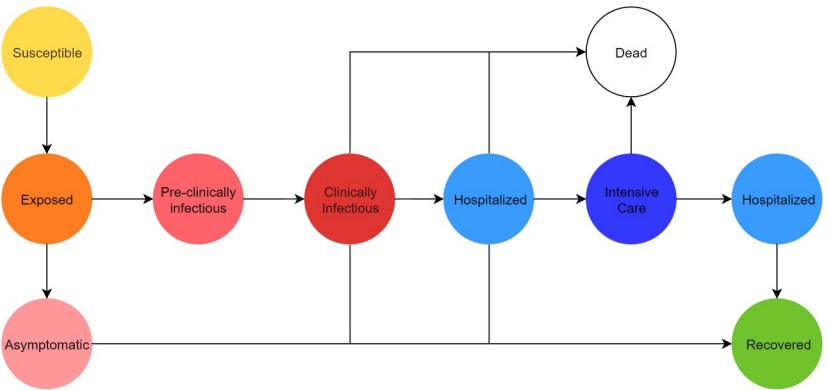

**Fig 8. Disease state diagram.** The directed graph of health states.

We do not assume limits on hospital and intensive care capacity, since we lack appropriate data. In particular, we have not tried to estimate the conditional probability of death given that the hospital or ICU is full.

We do not assume that time spent in a health state is geometrically distributed. Instead, we configure these durations according to the various distributions published in [89]. Denoting by $\Gamma(\alpha, \beta)$ the Gamma distribution with shape parameter $\alpha$ and scale parameter $\beta$ and by $U(a, b)$ the uniform distribution on the integers $\{a, \ldots, b\}$, the distributions of time agents spent in each health state for each pathway are then configured as in Table 5, in which the first and last states are ignored.

We assume that a number of agents are initially infected with the virus. These agents are selected at random from among the resident population. Agents move between locations, and should a susceptible agent be in the same location as an infectious agent during the same 10 minute time interval, then with a certain probability a new infection will occur. In particular, for a given time and for each location $l$, denote by $n_{\mathrm{sym}}(l)$ and $n_{\mathrm{asym}}(l)$ the numbers of symptomatic (**C**, **H** or **I**) and asymptomatic (**A** or **P**) agents at location $l$. Then, in the absence of interventions, denoting by $p$ the transmission probability and by $\alpha \in [0, 1]$ the reduction in transmission probability for asymptomatic agents, we assume that the probability $p(l)$ that a susceptible agent at location $l$ is infected is

$$p(l) = 1 - (1 - p)^{n_{\mathrm{sym}}(l)}(1 - \alpha p)^{n_{\mathrm{asym}}(l)} \tag{1}$$

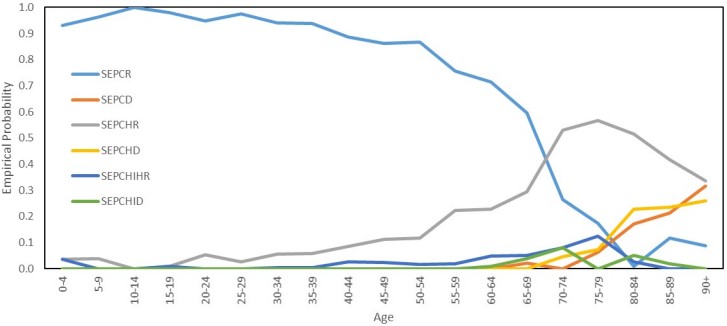

**Fig 9. Probabilities of symptomatic disease pathways.** The age-dependent distribution of symptomatic disease pathways appearing in our model.

**Table 5. Duration of time spent in each health state, in days, ignoring the first and last state in each pathway.**

| Pathway | Durations |
|---|---|
| SEAR | $\Gamma(4, 0.75) \rightarrow \Gamma(4, 1.250)$ |
| SEPCR | $\Gamma(4, 0.75) \rightarrow \Gamma(4, 0.525) \rightarrow \Gamma(4, 0.725)$ |
| SEPCD | $\Gamma(4, 0.75) \rightarrow \Gamma(4, 0.525) \rightarrow U(10, 15)$ |
| SEPCHR | $\Gamma(4, 0.75) \rightarrow \Gamma(4, 0.525) \rightarrow U(5, 9) \rightarrow U(8, 21)$ |
| SEPCHD | $\Gamma(4, 0.75) \rightarrow \Gamma(4, 0.525) \rightarrow U(5, 9) \rightarrow U(5, 16)$ |
| SEPCHIHR | $\Gamma(4, 0.75) \rightarrow \Gamma(4, 0.525) \rightarrow U(5, 9) \rightarrow U(3, 5) \rightarrow U(2, 12) \rightarrow U(2, 4)$ |
| SEPCHID | $\Gamma(4, 0.75) \rightarrow \Gamma(4, 0.525) \rightarrow U(5, 9) \rightarrow U(3, 5) \rightarrow U(7, 10)$ |

implying that the number of new infections at $l$, at the given time, is a Binomial random variable with distribution $B(n_{sus}(l), p(l))$, where $n_{sus}(l)$ denotes the number of susceptible agents at location $l$. If an agent is infected, then their health state moves from **Susceptible** to **Exposed**. We assume that $\alpha = 0.55$, meaning that asymptomatic and pre-clinically infectious agents are only 55% as infectious as the symptomatic infectious agents [90]. In the absence of personal protective measures, we assume that the transmission probability $p$ is uniform across location types, except outdoors (including construction sites) and in the border countries, where it is set to zero.

## Interventions

We assume perfect compliance with all interventions, except face masks and vaccination. In the case of face masks, we assume that low face mask availability results in some agents not wearing the masks, while for vaccination we will consider the possibility that agents refuse the vaccine. Agents in health state **Hospitalized** or **Intensive Care** are placed in a hospital, while agents in state **Dead** are moved to the cemetery.

**Testing.** We split testing into a number of sub-processes. Firstly, there is a process representing large scale testing, which on certain dates distributes large numbers of test invitations. While this process is based on the system of large scale testing used in Luxembourg, where test invitations were not distributed completely randomly, we assume for simplicity that they are. We assume that there is a delay between agents receiving an invitation for large scale testing and the booking of the test. We assume this delay is distributed randomly as in Fig 10, the data for this having being collected by General Inspectorate of Social Security in Luxembourg in 2020.

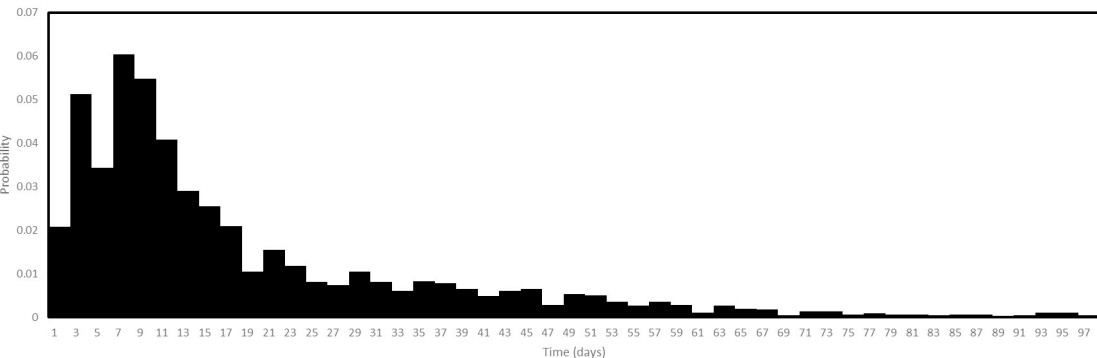

**Fig 10. Test booking delay.** The probability distribution of time between an agent receiving an invitation for large scale testing and the agent booking the test.

Secondly, there is a process representing prescription testing, in which agents book a test one day after having developed symptoms. There is then a test booking system, which handles these booking requests. We assume that if an agent has symptoms then the test takes place two days after the booking, while if an agent does not have symptoms then, given a lesser sense of urgency, it takes place four days after the booking. A laboratory process then performs the tests, returning results after two days with a 1% probability of a false negative. In addition, we assume that the laboratory is only able to perform a limited number of tests per day, the exact capacity being scenario-specific.

**Contact tracing.**   At the end of each day, any agent newly testing positive will have their contacts selected for testing and quarantine. Contacts are in this case defined to be those other agents who share a location with the given agent when performing the activities **House**, **Work** or **School**. These are the regular contacts of the agent, who the agent could be expected to identify through a manual search. Moreover, each day we limit the number of newly tested agents who are able to have their contacts traced, to model a limited scenario-specific capacity within the contact tracing system.

**Quarantining.**   Quarantining directs agents to perform all activities at their home location, for a default period of 14 days. Agents located in **Hospital** or the **Cemetery** are exempt from this directive. Should an agent obtain a negative test during their period of quarantine, then agents are able to leave quarantine restrictions after an additional 2 days.

**Face masks.**   According to the preprint [91], the effect of face masks can be modelled in terms of the mask transmission rate and mask absorption rate, which denote the proportions of viruses that are stopped by the mask during exhaling versus inhaling, respectively. This was also the approach taken in [24]. We assume these proportions are equal, this value being denoted $r \in [0, 1]$. Then, given a susceptible agent $i$ and an infectious agent $j$, both in location $l$ at the same time, and denoting by $q$ the probability of an agent wearing a mask, it follows that the probability that $j$ infects $i$ is reduced by a factor

$$(1 - r)^2 q^2 + 2q(1 - r)(1 - q) + (1 - q)^2 = (1 - rq)^2.$$

Moreover $q$ can be expressed as the probability that an agent wears a mask given that the agent has a mask, multiplied by the probability that the agent has a mask. Following the authors of [91], we set $r = 0.7$, with $q$ scenario-specific.

**Curfew.**   On 26th October 2020, an 11pm-6am curfew was imposed in Luxembourg. In our implementation, a curfew directs agents home between these hours unless they are located in **Hospital** or the **Cemetery**. While this implementation captures the essence of the curfew, it does not capture how a curfew affects the behaviour of individuals earlier in the evening. On the one hand, individuals might cancel plans altogether to avoid breaking the curfew, while on the other they might simply perform the same activities but earlier. In this study, we do not consider such effects.

**Location closure.**   Location closures set locations of certain types inaccessible to agents between certain dates, with agents attempting to access such locations being instead directed home. Location closures can be used to model lockdowns, school closures or closures of specific economic sectors. In the special case of care home closures, we allow agents access if they work at the care home, meaning that in this case only visits are prohibited, while in the special case of shops we permit each shop to stay open with a certain probability, since in reality not all shops close during a lockdown. Typically shops selling food, drink or fuel will remain open. According to [85], approximately 72% of shops in Luxembourg do not sell either food, drink or fuel.

**Vaccination.** Recall that at each tick of the simulation clock, in the absence of interventions, an agent $i$ in health state **Susceptible** moves to state **Exposed** with probability $p(l)$ given by Eq 1, where $l$ denotes the current location of the agent. Our implementation of vaccination associates to each agent $i$ a quantity $v_i \in \{0, 1\}$ and replaces the probability $p(l)$ by $(1 - v_i)p(l)$. If $v_i = 1$ then agent $i$ is protected against infection, meaning that the health of agent $i$ is protected against entering the state **Exposed**. Agents for whom $v_i = 1$ are therefore unable to catch the virus, however agents who previously caught the virus, and are not yet either recovered or dead, are still able to transmit it. When an agent $i$ is administered with a dose of the vaccine, with a certain probability $v_i$ is set equal to 1.

We assume the vaccine is administered in a two-dose format, with a fixed time between doses. We assume that the first and second doses protect the recipient with probabilities $p_1$ and $p_2$, respectively. Therefore, for an agent $i$, we have $v_i = 1$ with probability $p_1$ after the first dose and with probability $p_1 + (1 - p_1)p_2$ after the second dose. For example, if the latter probability is set equal to 0.557, with $p_1$ set equal to 0.463, following [10], then we must set $p_2 =$ 0.175. We use these probabilities to represent vaccine efficacy, meaning therefore efficacy against infection. We assume, for simplicity, that agents are either totally protected against infection or not at all.

Moreover we assume that everyone who receives a first dose later receives the second dose. We assume that only a certain number of first doses of the vaccine can be administered each day and that agents are vaccinated in a particular order. The default scheme starts with care home residents and care home workers, followed by hospital workers, followed by everyone else, with each of these categories ordered by age, down to a minimum age of 16. We also assume that agents refuse vaccination with a certain probability, depending on their age. Such hesitancy is realized in our model by randomly selecting agents and having these agents refuse the vaccine at the moment of injection.

## Model evaluation

We will configure the model over the 129 day period from February 23rd 2020 to 30th June 2020, covering the first wave of cases in Luxembourg. The numbers against which we are calibrating are small, since data on the early stages of the COVID-19 epidemic in Luxembourg were limited, with very little testing taking place. However, over a longer time horizon other uncertainties would increase, due to factors not represented in our model becoming increasingly influential, such as loss of immunity, mutations and the impact of long distance travel.

## Parametrization of interventions

We must calibrate the interventions to reproduce the sequence of interventions that occurred in Luxembourg during the first four months of the epidemic. This is achieved using a scheduling system, which allows the interventions listed in the previous section to be enabled or disabled, and their parameters updated, on selected dates.

**Testing.** We assume that the capacity of the test laboratory is limited by the 7-day rolling average of the total number of tests recorded each day in Luxembourg. These daily totals, together with the trendline, are plotted below in Fig 11, between 1st March 2020 and 30th June 2020.

The parametrization of large scale testing invitations is illustrated in Fig 12. This shows, approximately, the dates on which test invitations were sent in Luxembourg and the numbers of invitations sent on those dates. Recall that our agents respond to these invitations with a random delay.

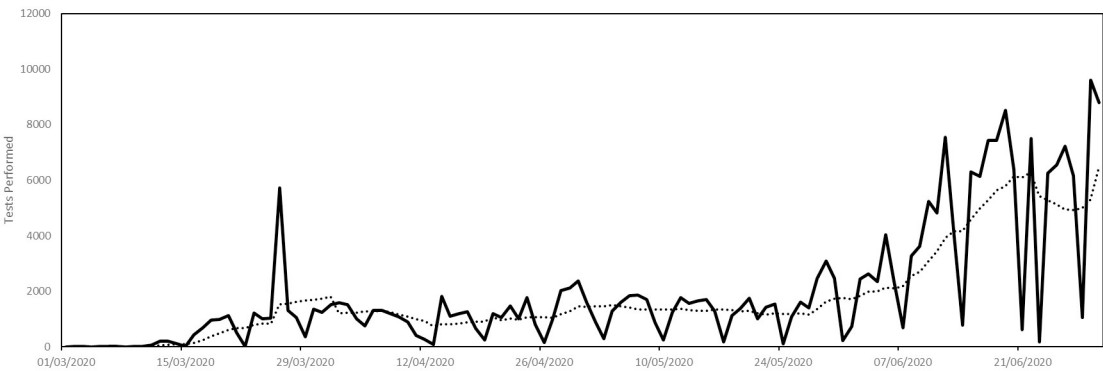

**Fig 11. Parametrization of laboratory testing capacity.** The capacity is given by the trend line.

**Contact tracing.** We assume that contract tracing starts on 20th April 2020 with a capacity of 100. This means that as many as 100 agents testing positive each day can have their regular contacts traced. The capacity of the contract tracing system in Luxembourg subsequently increased, but not until much later.

**Face masks.** We assume that agents do not initially have access to face masks, the probability that they do increasing to 0.8 on 20th April 2020 and from 0.8 to 1.0 on 11th May 2020. We assume that the probability of a mask being worn, given that masks are available, depends of the type of location. We assume this probability is 0.0 inside houses and cars and 1.0 inside public transport, shops, medical clinics, hotels, places of worship and museums and zoos. Elsewhere we assume this probability is 0.2. We assume moreover that in hospitals and medical clinics face masks are always available and that they are always worn.

**Location closure.** Below, in Fig 13, we plot a time line indicating when locations of various types are assumed inaccessible during our validation simulations. The category General Work appearing in Fig 13 refers to location types listed in Table 2, except **Primary Schools**, **Secondary Schools**, **Construction** and **Entertainment**, which are listed separately. Leisure refers to locations of type **Indoor Sport**, **Cinema or Theatre**, **Museum or Zoo** and **Restaurants**. Closure of locations of type **House** or **Care Home** means that agents are unable to access these locations while preforming the activity **Visit**.

In addition, we assume that 72% of shops, chosen at random, close from 15th March 2020 to 11th May 2020, since according to [85] approximately this percentage of shops in Luxembourg do not sell either food, drink or fuel and were therefore subject to such restrictions.

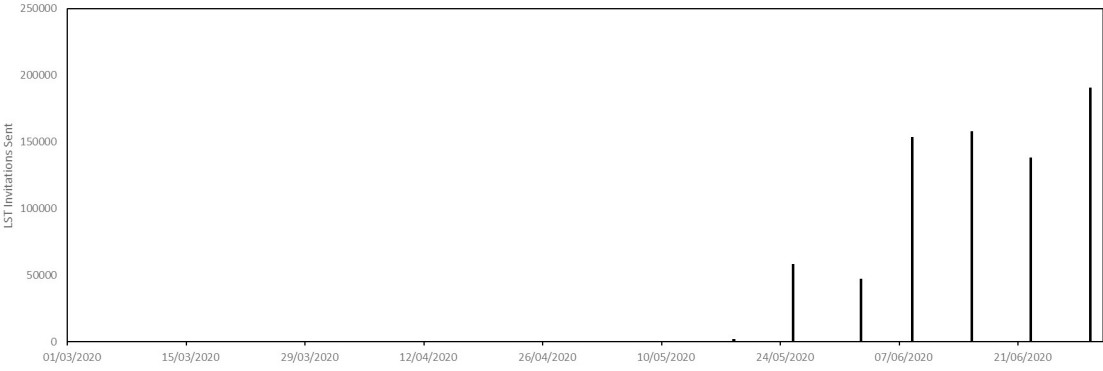

**Fig 12. Parametrization of large scale testing schedule.**

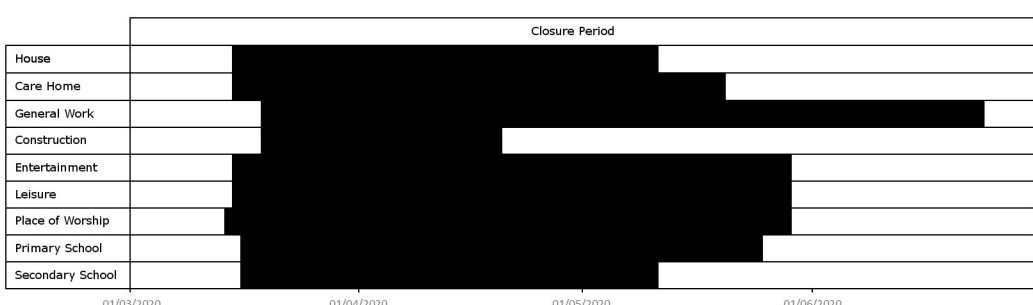

**Fig 13. Schedule of location closures.** A partial timeline of location closures in Luxembourg, from 1st March 2020 to 30th June 2020. The lockdown began on roughly 15th March 2020.

## Validation

With the interventions and other components configured, it remains to calibrate the initial infection count, transmission probability and the age-dependent probabilities of asymptomatic infection.

Initial conditions are set by randomly selecting a number of residents, and setting their initial health state to be the first infectious state appearing in their assigned disease pathway. This means, for example, that if a selected agent has disease pathway **SEPCR**, then their starting health state will be **Pre-clinically Infectious**. We assume there were 320 cases in Luxembourg on 23rd February 2020.

Infected agents are either symptomatic or asymptomatic. During initialization, we assign agents the asymptomatic pathway **SEAR** with a probability that depends on their age. As a starting point for such probabilities, we take the numbers reported in [92]. Then, for each agent of age $a$, we have a probability $A(a)$ that the agent will be assigned **SEAR**. To account for uncertainty in these probabilities, we introduce a parameter $s \in [0, 1]$ to interpolate between these probabilities and the extreme case in which all agents are asymptomatic. Given an agent of age $a$, the probability that they are assigned **SEAR** is then $A(a)(1 - s) + s$. The parameter $s$ gives us control over the probabilities of hospitalization and death, without disrupting the distributions visualized in Fig 9. In Fig 14, we plot the age-dependent asymptomatic probabilities for the three values $s = 0$, $s = 0.2$ and $s = 0.4$. While $s = 0$ corresponds exactly to the probabilities quoted in [92], our simulations suggest these probabilities are too low, and therefore our validation process will consider only $s = 0.2$ and $s = 0.4$.

For the transmission probability $p$, appearing in Eq 1, we consider the three values $p = 0.00015$, $p = 0.00025$ and $p = 0.00035$. Table 6 then shows the range of values of the pair ($s$,

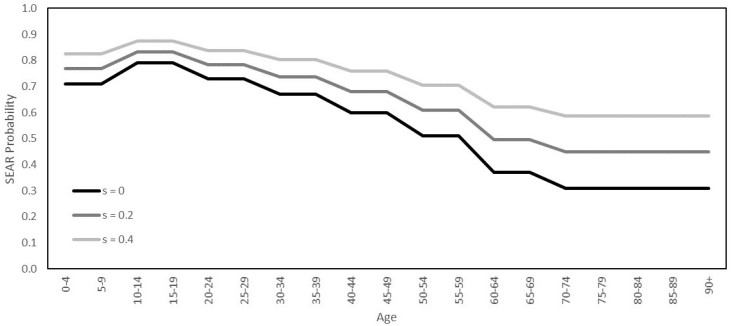

**Fig 14. Age-dependent asymptomatic probabilities.** The probabilities of an agent being assigned the pathway **SEAR**.

**Table 6. Pairs (*s*, *p*) in the small grid search.** Here *s* is the parameter that controls the probability of being asymptomatic, while *p* is the transmission probability.

| | |
|---|---|
| $s = 0.2$, $p = 0.00015$ | $s = 0.4$, $p = 0.00015$ |
| $s = 0.2$, $p = 0.00025$ | $s = 0.4$, $p = 0.00025$ |
| $s = 0.2$, $p = 0.00035$ | $s = 0.4$, $p = 0.00035$ |

*p*) over which we will perform a small grid search. A more sophisticated validation process is not possible at the present time, due to the computational burden of the agent-based model.

We will perform all simulations at 0.25 scale, meaning that all relevant quantities are reduced to a quarter of their full size. Such quantities include population size, the number of locations and various quantities relating to the interventions, such as testing and contact tracing capacity. We then rescale the output to full size, multiplying by 4 all relevant quantitative output. At 0.25 scale our simulations each took around 5 hours.

We performed 10 simulations for each pair of parameter values appearing in Table 6. In Fig 15, we plot the corresponding numbers of resident deaths and hospitalizations for each simulation (grey and pink, respectively), together with their averages (solid black and red, respectively) and the numbers of deaths and hospitalizations recorded in Luxembourg over the same time period (dotted black and red, respectively). We calculate the number of hospitalizations in a simulation by adding the numbers of agents whose health state is either **Hospitalized** or **Intensive Care**.

The pair *s* = 0.4, *p* = 0.00035 produces the closest fit, so these are the parameters that will be used in all subsequent simulations. In Fig 16, we plot the average numbers, across the 10 simulations corresponding to the pair *s* = 0.4, *p* = 0.00035, of agents in the health states **Exposed**, **Asymptomatic**, **Pre-clinically Infectious**, **Clinically Infectious**, **Hospitalized**, **Intensive Care** and **Dead**.

Fig 16 shows how in our model most new exposures occur on weekdays during working hours, with more towards the beginning of the week than the end. The impact of daily and weekly cycles, resulting from the activity model and the time use data, is clearly visible.

## Sensitivity analysis

We performed a simple univariate sensitivity analysis for the unknown parameters *s* and *p*, for the baseline scenario established above. Recall that *p* is the transmission probability while *s* is the parameter that controls the probability of being asymptomatic. We measured the sensitivity by estimating the partial derivatives, with respect to these two variables, of the total dead after 129 days. We also estimated the partial derivative with respect to the total number of care homes, since this is a quantity that we expect plays a significant role in determining the final outcome of the epidemic. We estimated the number of care homes in Luxembourg at 52, and now consider a variation in this number over the interval [10, 100]. We vary *s* over the interval [0.0, 0.8], with *s* = 0.4 corresponding to the baseline scenario, and we vary *p* over the interval [0.000175, 0.000525], with *p* = 0.000350 corresponding to the baseline scenario. In Fig 17 we plot the variations in *s* and *p* against the corresponding model output.

We see from Fig 17 that the total number of deaths is not highly sensitive to independent variations of the parameters *s* and *p*. The relationships are, in fact, approximately affine linear. Therefore, since the primary objective of this article is to compare interventions against the baseline scenario, with the interventions being implemented on top of the baseline model in a very natural way, we do not expect our final conclusions to be highly sensitive to small

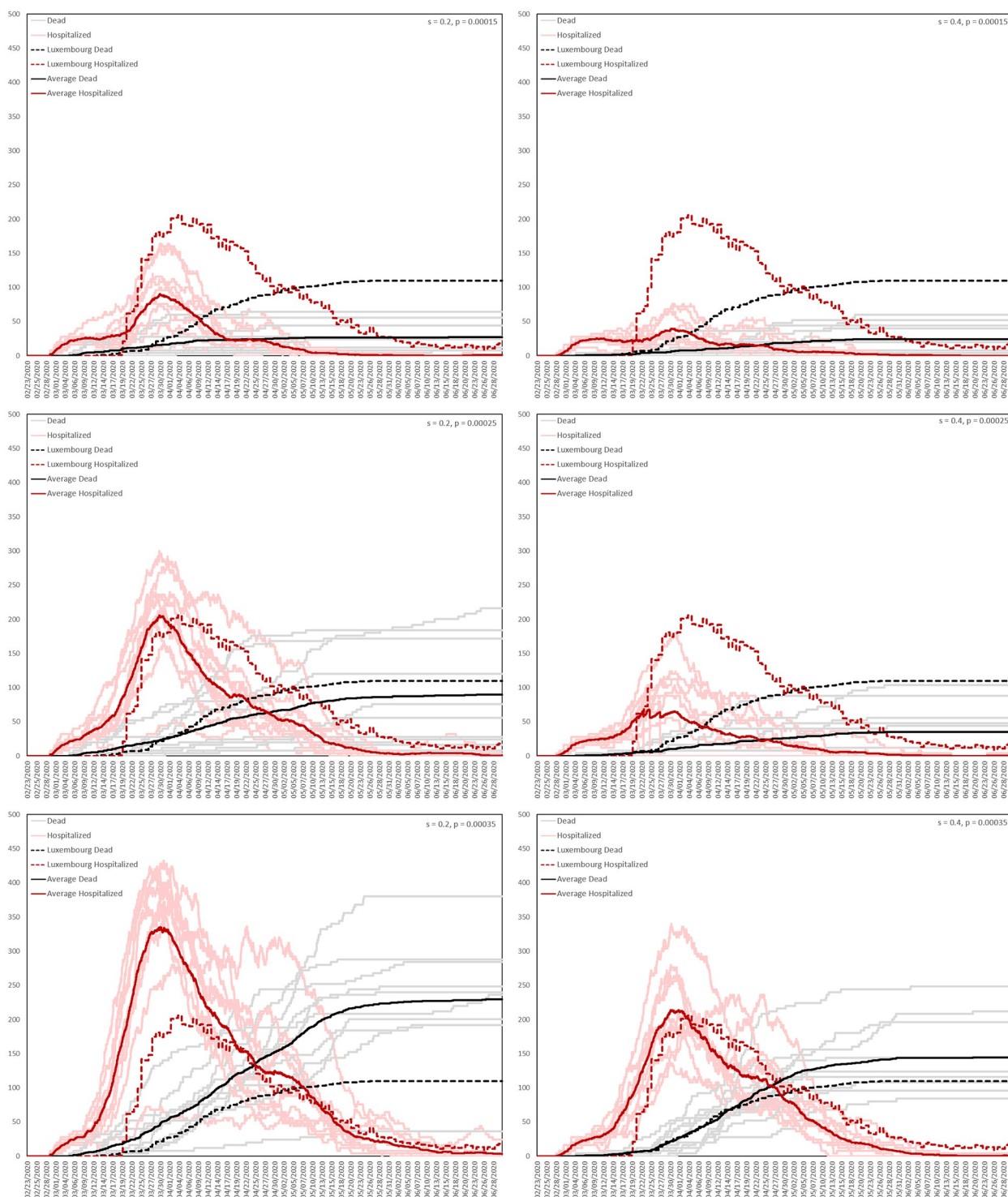

**Fig 15. Results of the grid search.**

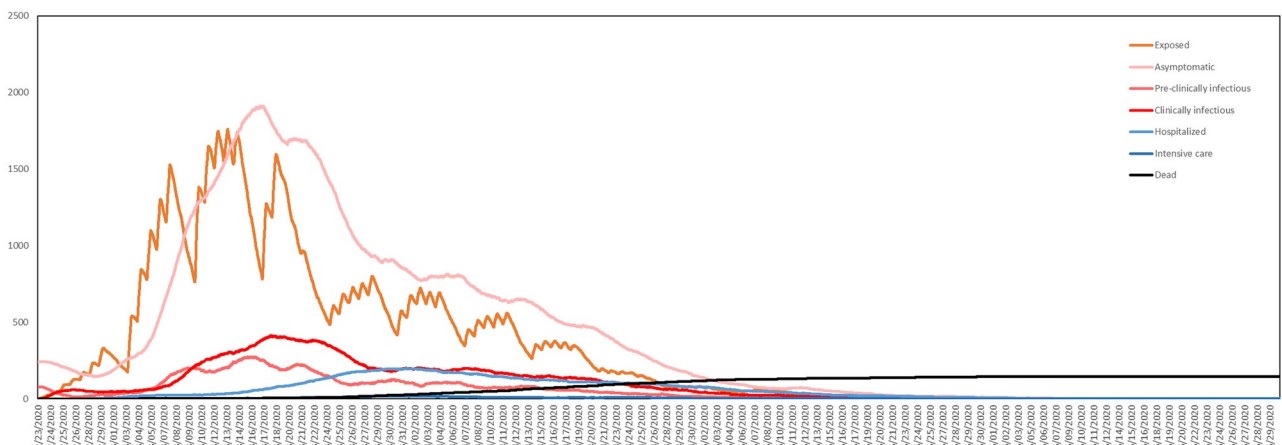

**Fig 16. Numbers of resident agents in each health state.** Averaged over 10 simulations with $s = 0.4$, $p = 0.00035$.

variations of these parameters. In Fig 18, we plot the variation with respect to the number of care homes.

Fig 18 shows that the total number of dead in our model is also not highly sensitive to the number of care homes, the dependence being again affine linear. Nonetheless, the figure shows that in our model, increasing the number of care homes increases the total number of dead.

## Results

We now consider a number of different scenarios, using the parametrization $s = 0.4$ and $p = 0.00035$, with ten simulations performed for each scenario. Each simulation runs over the same 129 day interval, but with a different random seed. For scenarios involving interventions, we suppose the interventions activate after exactly 3 weeks and continue until the end of the simulation.

### Baseline

In the baseline scenario, no interventions are active, meaning that agent behaviour does not change in response to the epidemic. In this case, we compare the output of our agent-based

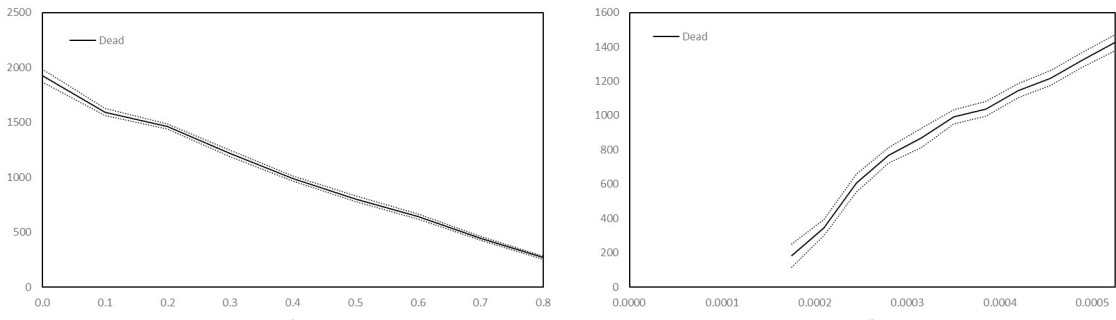

**Fig 17. Number of dead resident agents after 129 days versus _s_ and _p_.** The solid curves plot the mean total deaths and the dotted curves the two-sided 95% critical region, calculated using 180 simulations for the variations in $s$ and 100 other simulations for the variations in $p$.

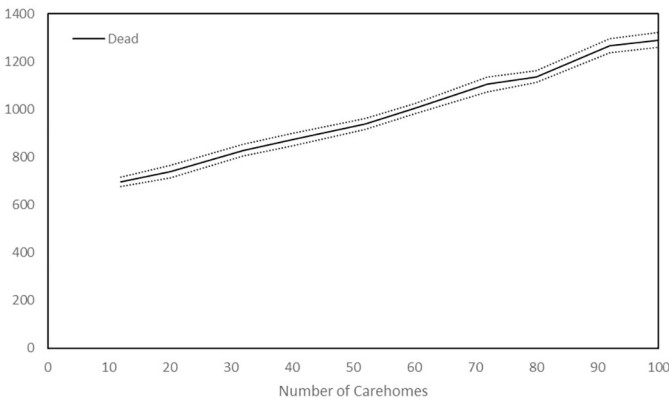

**Fig 18. Number of dead resident agents after 129 days versus number of care homes.** The solid curve plots the mean total deaths, the dotted curves plot the two-sided 95% critical region, calculated using 400 simulations.

model to that of the equation-based SEIR model. Table 7 shows how the health states of the SEIR model correspond to those of the agent-based model (ABM).

For the agent-based model, we plot in Fig 19 the numbers Exposed and Infected in each of 100 simulations of the baseline scenario, together with the corresponding means. Similarly, in Fig 20 we plot the numbers **Dead**.

The baseline scenario results, on average, in approximately 985 deaths among the resident population of Luxembourg. This compares to a recorded 110 COVID-19 deaths over the same period ending 30th June 2020, when numerous interventions against COVID-19 were active in Luxembourg.

The SEIR model is given by system of ordinary differential equations

$$\frac{d}{dt}S = -\beta\frac{SI}{N}$$

$$\frac{d}{dt}E = \beta\frac{SI}{N} - \alpha E$$

$$\frac{d}{dt}I = \alpha E - \gamma I$$

$$\frac{d}{dt}R = \gamma I$$

**Table 7. Equivalence of health states between the SEIR model and ABM.**

| SEIR | ABM |
|---|---|
| Susceptible | Susceptible |
| Exposed | Exposed |
| Infected | Asymptomatic |
| | Pre-clinically Infectious |
| | Clinically Infectious |
| | Hospitalized |
| | Intensive Care |
| Removed | Recovered |
| | Dead |

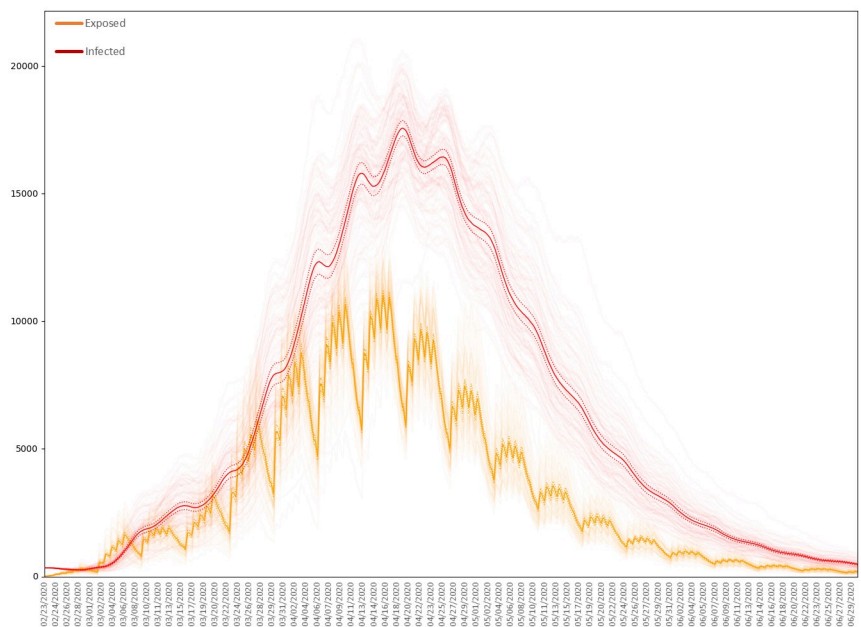

**Fig 19. Exposed and infected residents in the baseline scenario.** 100 simulations of the baseline scenario with $s = 0.4$ and $p = 0.00035$. The dark curves plots the means, the dotted curves the two-sided 95% critical regions for the means.

with initial conditions

$$(S(0), E(0), I(0), R(0)) = (625920, 0, 320, 0).$$

For such a model it is assumed that the incubation and infectious periods are exponentially distributed with mean durations $\alpha^{-1}$ and $\gamma^{-1}$, respectively. We set

$$\alpha^{-1} = 6.0512 \text{ days}, \quad \gamma^{-1} = 3.0020 \text{ days}$$

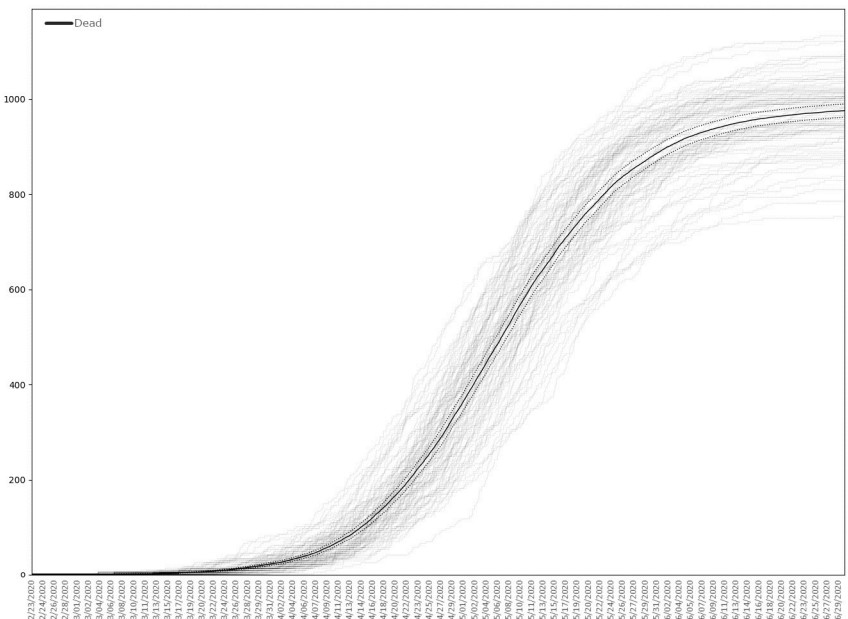

**Fig 20. Dead residents in the baseline scenario.** 100 simulations of the baseline scenario with $s = 0.4$ and $p = 0.00035$. The dark curve plots the mean, the dotted curves the two-sided 95% critical regions for the mean.

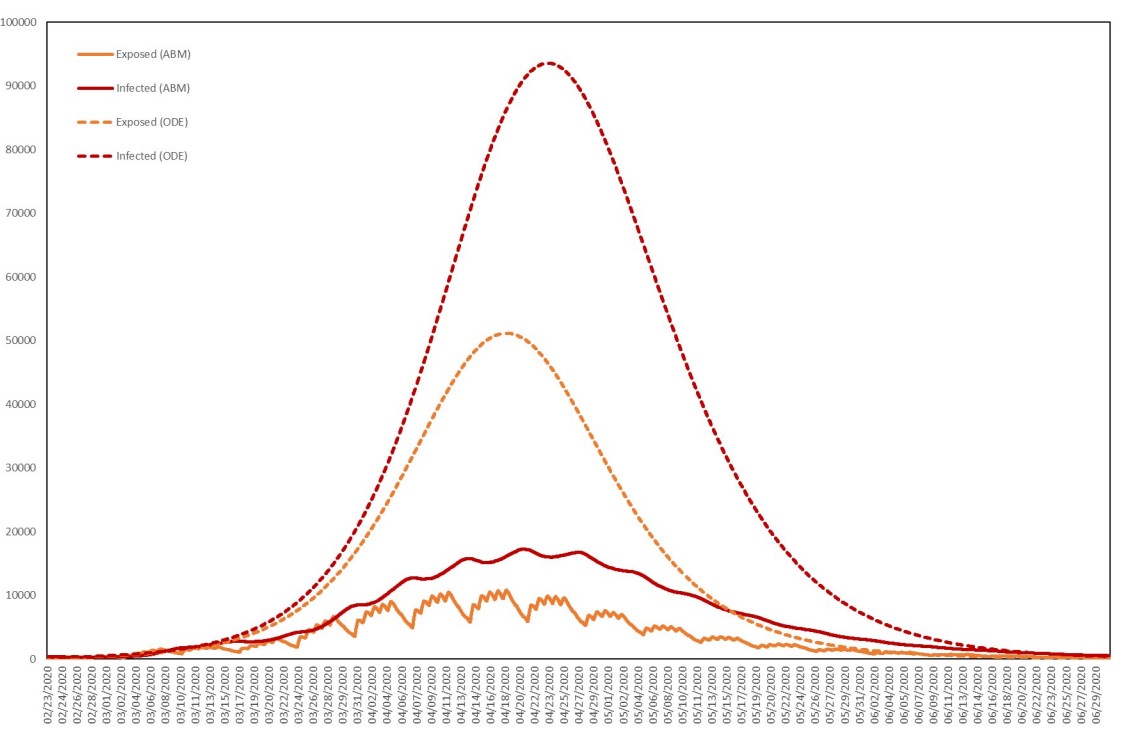

**Fig 21. Numbers of resident agents exposed and infected.** The average numbers exposed and infected in the agent-based model (ABM) with $p = 0.00035$ and in the SEIR ordinary differential equation (ODE) model with $R_0 = 2.45$.

since these are the average incubation and infectious periods among residents in the agent-based model. The basic reproduction number of the SEIR model, denoted $R_0$, is given by

$$R_0 = \frac{\beta}{\gamma}.$$

Choosing $R_0$ therefore determines $\beta$. To be precise, $\beta$ is the average number of contacts per person per day, multiplied by the probability of disease transmission in a contact between a susceptible and an infectious individual. We observe that setting $R_0 = 2.45$, and therefore $\beta = 0.4049$ days$^{-1}$, yields a solution that peaks at roughly the same time as the epidemic produced by the agent-based model with $p = 0.00035$. For the two models, we plot the numbers Exposed and Infected in Fig 21.

As shown in Table 8, the agent-based model (ABM) predicts considerably fewer cases than the SEIR model (where by a case we mean any agent either exposed or infected). This highlights the impact of heterogeneity and stochasticity [93], with spatial clustering limiting the reach of infected individuals and daily and weekly routines fragmenting the contact network at night and over weekends. For Table 8 we have also calculated the $t$-statistic of the difference, using the sample standard deviation of the realizations of the ABM used to calculate the

**Table 8. Average cumulative cases in the resident population, for the SEIR model and ABM, with $R_0 = 2.45$ and $p = 0.00035$, respectively, for a total resident population of 626,240, together with the $t$-statistic of the difference.**

|  | SEIR | ABM | $t$-statistic |
|---|---|---|---|
| Total cases | 554,673 (87%) | 143,162 (23%) | 278.4515 |

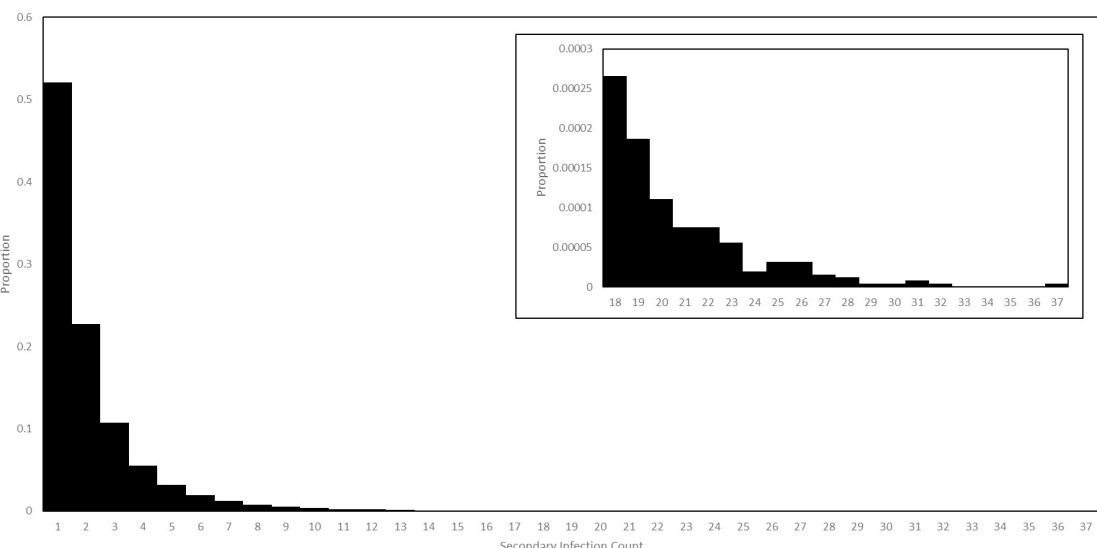

**Fig 22. Secondary infection counts in the baseline scenario.** The probability distribution of secondary infections counts among those agents who caused at least one, in a simulation of the baseline scenario. The tail of the distribution is displayed in the inset plot, on a different scale.

average, emphasizing that the output of the ABM is indeed significantly different from that of the SEIR model.

Our model records not only the numbers of agents in each health state at each time, but also data on transmission events. In the baseline scenario, we found that approximately 12% of all agents caused secondary infections. Among those who did, the probability distribution of the number of secondary infections is displayed in Fig 22. While the majority of agents who caused secondary infections caused only 1 or 2, a few caused as many as 37, with these agents therefore playing the role of super spreaders. The majority of infections caused by these super spreaders occurred at work. Among all agents, the average number of secondary infections was 0.27 while among only those who caused at least one secondary infection the average was 2.14.

We are also able to calculate generation times. In the baseline scenario, among all agents who caused secondary infections, the maximum generation time was 44680 minutes, or approximately 31 days, while the mean was 7154 minutes, or approximately 5 days. We plot the full probability distribution in Fig 23, observing that it concentrates around multiples of 24 hours after infection.

While in this subsection we have sought to highlight the differences between our agent-based model and the equation-based SEIR model, the SEIR model referred to above is among the simplest of the compartmental models, and increasingly more detailed sets of equations would result in output progressively closer to that of the agent-based model. Indeed, an agent-based model can always be represented as a system of differential equations, however the number of such equations would be enormous.

## Individual interventions

Now that we have established the baseline scenario, we can simulate interventions and assess their impact by comparing to the baseline. We start with those interventions that act on the level of the individual. In particular, we consider different levels of prescription testing, large scale testing and contact tracing, looking at low, medium and high intensities. In each of these three scenarios the test booking and laboratory systems are active, together with the quarantine

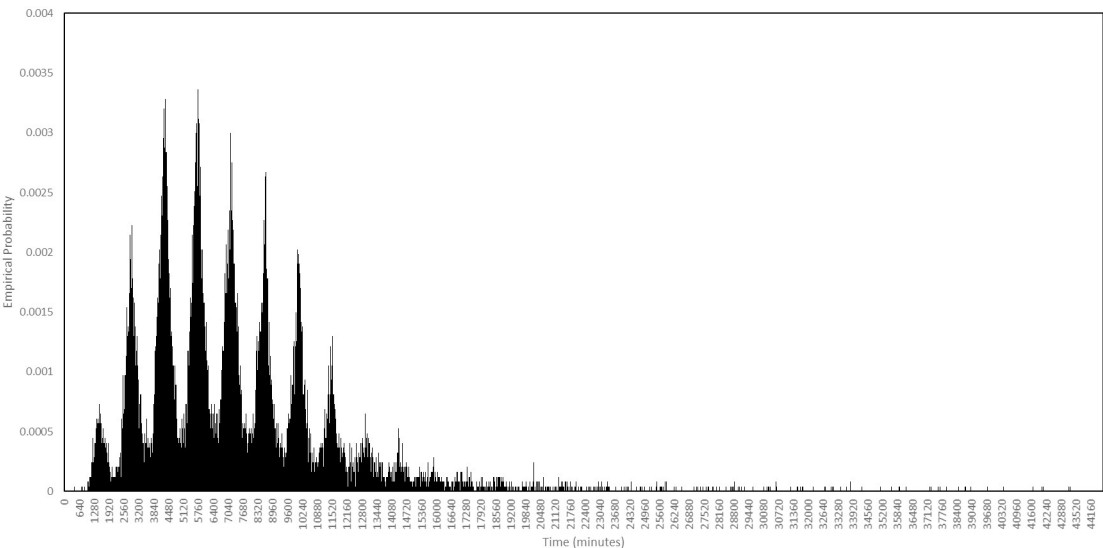

**Fig 23. Generation times in the baseline scenario.** Among those agents who passed on the virus, this is the probability distribution of the length of time between them being infected and them passing it on for the first time.

intervention. Recalling that the model represents a total resident population of 625920, the four scenarios are as follows:

- **Baseline**: Agents behave as normal.

- **Low**: A daily testing capacity of 1000, with 800 invitations for large scale testing sent each day, and a contact tracing capacity of 100.

- **Medium**: A daily testing capacity of 5000, with 4000 invitations for large scale testing sent each day, and a contact tracing capacity of 300.

- **High**: A daily testing capacity of 10000, with 8000 invitations for large scale testing sent each day, and a contact tracing capacity of 500.

Recall that the contact tracing capacity refers to the number of agents each day who, having tested positive, can have their regular contacts traced for testing and quarantine. For each scenario we performed ten simulations, with the interventions activating after exactly 3 weeks. The average numbers of cases and dead in the three scenarios are plotted in Fig 24, together with the baseline for comparison, where by a case we mean any agent either exposed or infected.

Based on these averages plotted in Fig 24, Table 9 summarizes the reductions in cases and deaths across the four scenarios, by the end of the 129-day simulation period.

Rapidly containing an outbreak using these testing and contact tracing systems is difficult, partly because of delays, for example between onset of symptoms and the publishing of test results, and partly because the quarantining of individuals takes place at home, where they might expose other residents. In particular, since care home residents spend almost all of their time in the came home anyway, quarantining them there has little effect in our model.

## Location interventions

In this subsection, we look at the impact of interventions that act on locations, rather than agents. We compare the following four scenarios, the last of which is hypothetical but similar in concept to proposals of other authors, for example [94]:

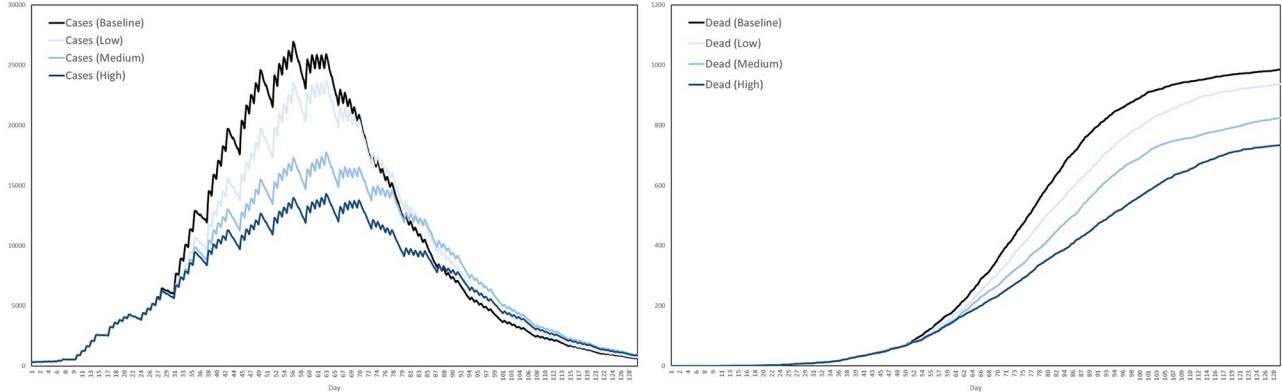

**Fig 24. Impact of testing and contact tracing.** The impact of a combined testing and contact tracing strategy, at three levels of intensity, on cases and deaths.

- **Baseline**: Agents behave as normal.

- **Curfew**: Agents must stay at home between 11pm and 6am unless they are in hospital.

- **Lockdown**: Agents must stay at home unless their destination is a hospital, a care home at which they work or one of the 38% of shops selling food, drink or fuel.

- **Targeted Lockdown**: Agents belonging to households containing at least one person over the age of 65 must stay at home, unless their destination is a hospital, care home or one of the 38% of shops selling food, drink or fuel.

In each case, the interventions activate 3 weeks into the simulation and continue until the end. We expected the curfew to have only a small impact. Indeed, according to the Luxembourg time use data, aggregated and displayed in Fig 4, we see that during the relevant hours the vast majority of people are typically at home anyway. Moreover, Fig 25 shows that mainly young people are out between these hours, except on weekday mornings when small numbers of adults of a broader range of ages are not at home, mostly commuting or starting work.

We expected the lockdown to have the biggest impact in reducing cases and deaths, while we expected the targeted lockdown to retain a substantial impact on deaths, but less so on cases. The targeted lockdown focusses on those agents most at risk of death, while allowing large numbers of other agents to continue with work and school. In Fig 26 we illustrate how cases and deaths compare across the four scenarios, where for each scenario we plot the average output of ten simulations, using the disease and transmission parameters of the baseline scenario and the same set of random seeds used elsewhere.

Based on these averages, Table 10 shows the corresponding reductions in cases and deaths versus the baseline scenario.

Table 10 shows that the impact of the lockdown is huge while that of the curfew is marginal, with the targeted lockdown somewhere in between (it could, however, be argued that our

**Table 9. Reductions in cases and deaths by testing and contract tracing, at low, medium and high intensities, versus the baseline scenario.**

|  | Baseline | Low | Medium | High |
|---|---|---|---|---|
| **Cases** | 0% | 6.5% | 18.2% | 30.5% |
| **Deaths** | 0% | 4.9% | 16.4% | 25.5% |

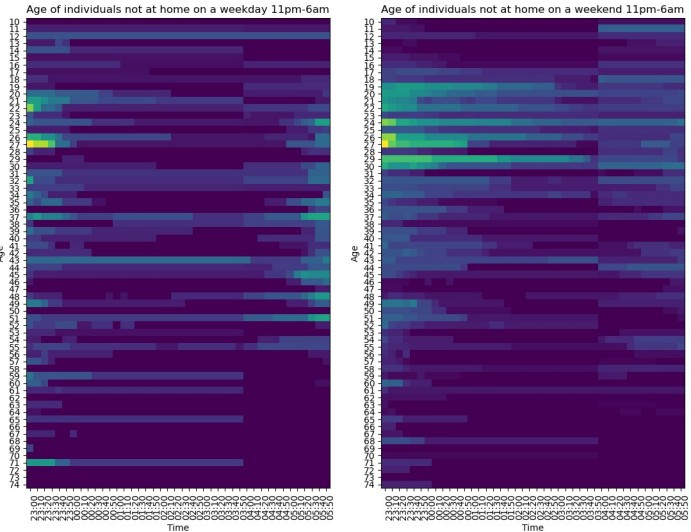

**Fig 25. Age distribution of people not at home from 11pm to 6am, under normal circumstances, Luxembourg 2014.** On the left, a weekday night and morning, for example Monday night and Tuesday morning. On the right, Saturday night and Sunday morning.

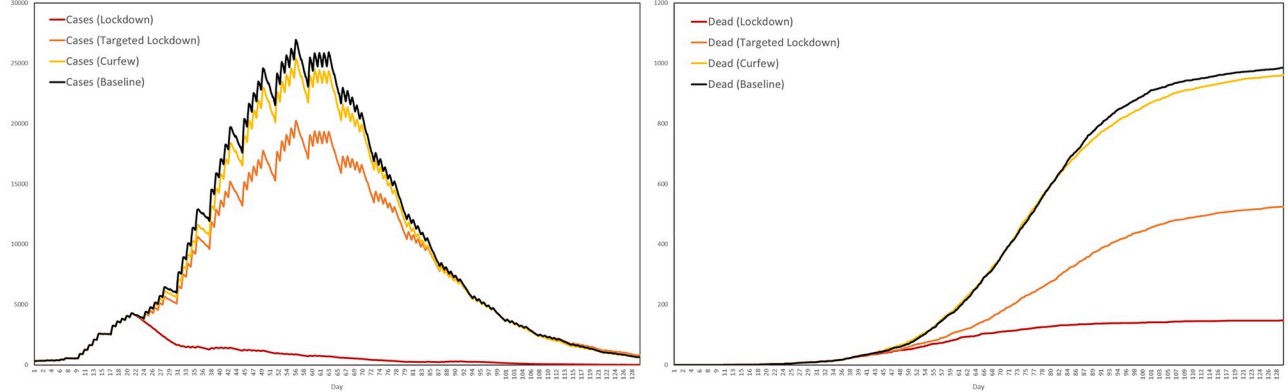

**Fig 26. The impacts of the curfew, targeted lockdown and lockdown on cases and deaths.** Averaged over 10 simulations and with the interventions activating on day 21.

estimate of the impact of the curfew is on the low side, since we do not consider the higher transmission levels present in bars and restaurants). To assess the disruption caused by these interventions, in Fig 27 we plot the distribution of agents across location types over the 2 week period from day 15 to day 28, illustrating the impact of these interventions on these distributions.

**Table 10. Reductions in cases and deaths by curfew, targeted lockdown and full lockdown, versus the baseline scenario.**

|  | Baseline | Curfew | Targeted Lockdown | Lockdown |
|---|---|---|---|---|
| **Cases** | 0% | 5.8% | 17.3% | 90.9% |
| **Deaths** | 0% | 2.4% | 46.7% | 85.1% |

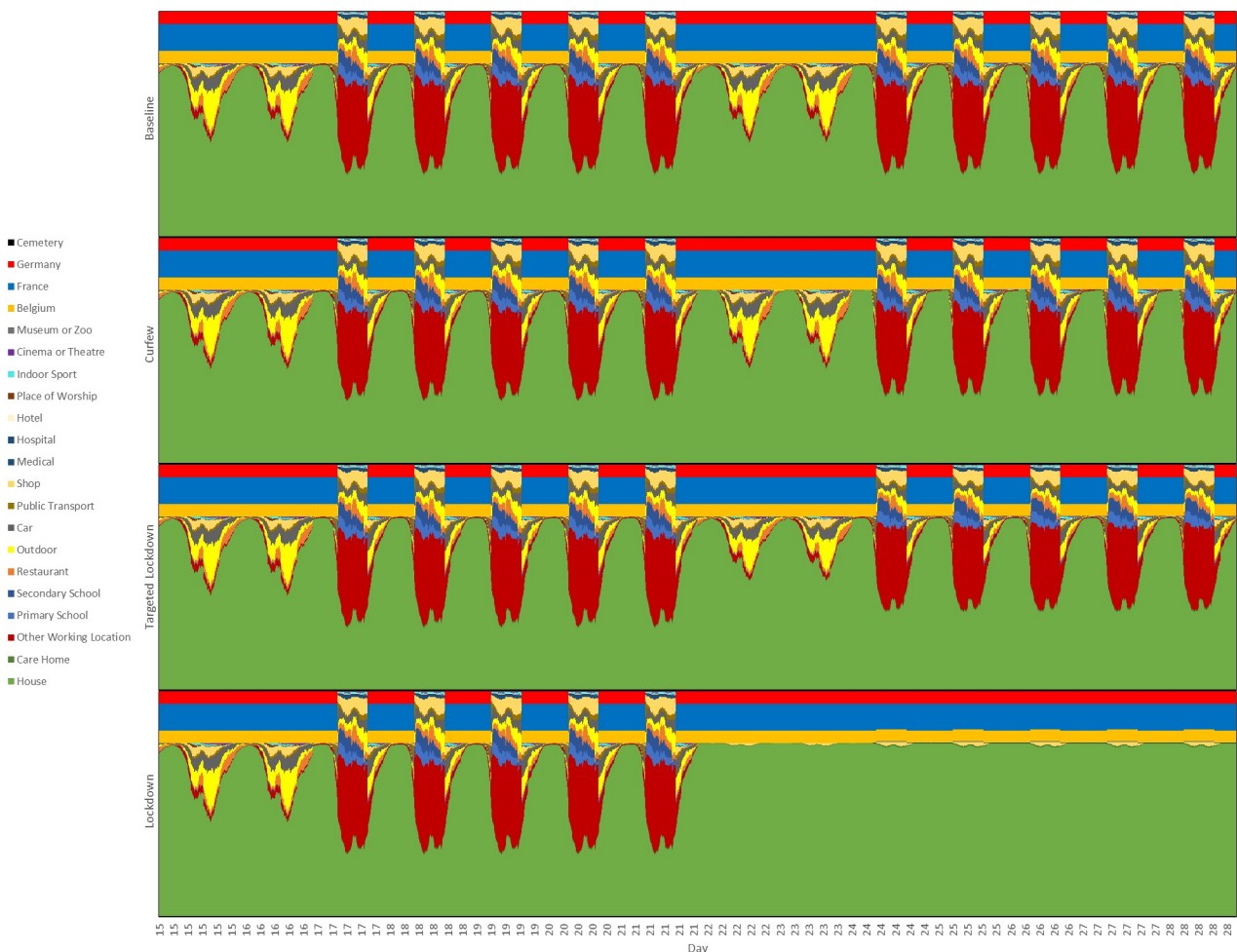

**Fig 27. Occupancy of different types of location under the baseline, curfew, targeted lockdown and lockdown scenarios.** The interventions activate 21 days into the simulation, and we therefore plot the 2 week period from day 15 to day 28. The data comes from four simulations, one for each scenario and all using the same random seed, to illustrate how the interventions impact the proportion of agents in locations of different types.

Fig 27 shows that the lockdown has a dramatic impact on the numbers of agents working and going to school, while the impact of the targeted lockdown on work and school is much milder. The impact of the curfew is also visible but very small. Much of what is achieved by the full lockdown is also achieved by the targeted lockdown, but with a considerably smaller economic and social cost. Such targeted lockdowns could in reality represent an alternative to a full lockdown.

Table 11 displays how, according to our model, these interventions impact the average number of secondary infections, calculated at the end of the simulation.

Table 11 shows that among agents who caused at least one secondary infection, these interventions have the effect of increasing the average, even while reducing the total number of

**Table 11. The impact of curfew, targeted lockdown and full lockdown on secondary infection counts.**

|  | Baseline | Curfew | Targeted Lockdown | Lockdown |
|---|---|---|---|---|
| Average number of secondary infections (among agents who caused at least one) | 2.14 | 2.15 | 2.17 | 2.42 |
| Average number of secondary infections (among all agents) | 0.27 | 0.26 | 0.23 | 0.02 |

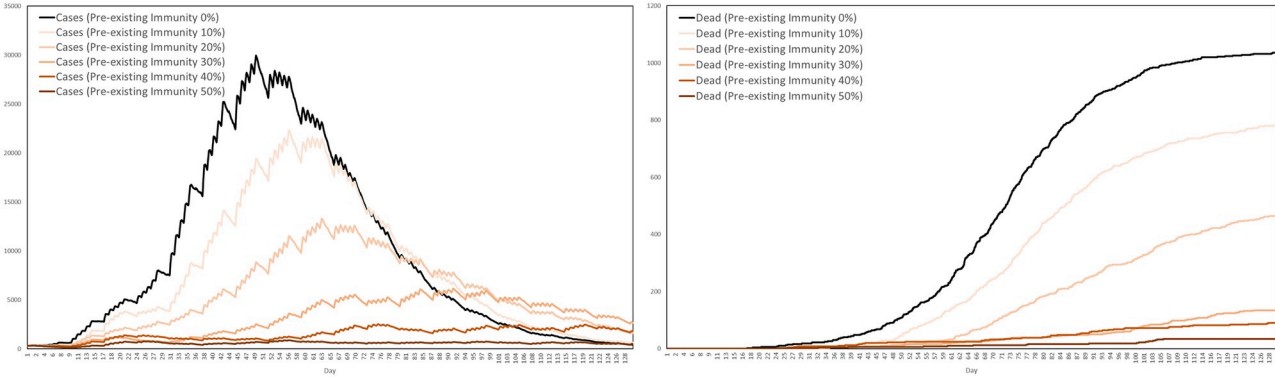

**Fig 28. Impact of pre-existing immunity on cases and deaths.** The impact of 0% up to 50% of the population having pre-existing immunity.

infections, highlighting the fact that when calculating averages, one must be careful with the choice of denominator.

## Vaccination

We now consider several scenarios relating to vaccination. We investigate herd immunity, efficacy, capacity, hesitancy and strategy.

**Herd immunity.** According to the World Health Organization [95]:

"'Herd immunity', also known as 'population immunity', is the indirect protection from an infectious disease that happens when a population is immune either through vaccination or immunity developed through previous infection."

Calculating the expression $1 − 1/R_0$ with $R_0 = 2.45$ implies a level of 59%. However, our model suggests that much lower levels of immunity provide the population with substantial protection against a future outbreak. Other studies have reached similar conclusions, for example [96]. We performed several simulations in which we assumed that a certain percentage of the population had pre-existing immunity. We selected these agents uniformly at random. In addition to two instances of the baseline scenario, where pre-existing immunity is 0%, we performed ten experiments in five pairs corresponding to levels of pre-exisiting immunity set at 10%, 20%, 30%, 40% and 50%. The simulations were otherwise parametrized as in the baseline scenario. For each pair, we averaged the two sets of outputs and the resulting numbers of cases and deaths are plotted in Fig 28.

Recalling that the baseline scenario results, on average, in around 23% of all agents infected, much lower than the 87% predicted by the SEIR model, we see from Fig 28 that pre-existing immunity of only 30% already has a dramatic impact on reducing total cases and deaths. This suggests that relatively low levels of coverage can help protect a population from future outbreaks. A different situation is the one in which an epidemic is already under way, with vaccination occurring in response to it. This is the situation that will be considered next. Also, with a view towards COVID-19 vaccination programmes starting in early 2021, such as in Luxembourg where a significant proportion of the population is already immune having been previously exposed to the disease, we will assume for all subsequent experiments that 10% of the population have pre-existing immunity. It was therefore necessary to perform ten additional simulations of the baseline scenario with 10% pre-existing immunity, with this new baseline being the one appearing in all subsequent figures.

**Efficacy.** Consider the situation where vaccination begins 3 weeks into the epidemic. We assume that vaccines are distributed first to care home residents and workers, followed by

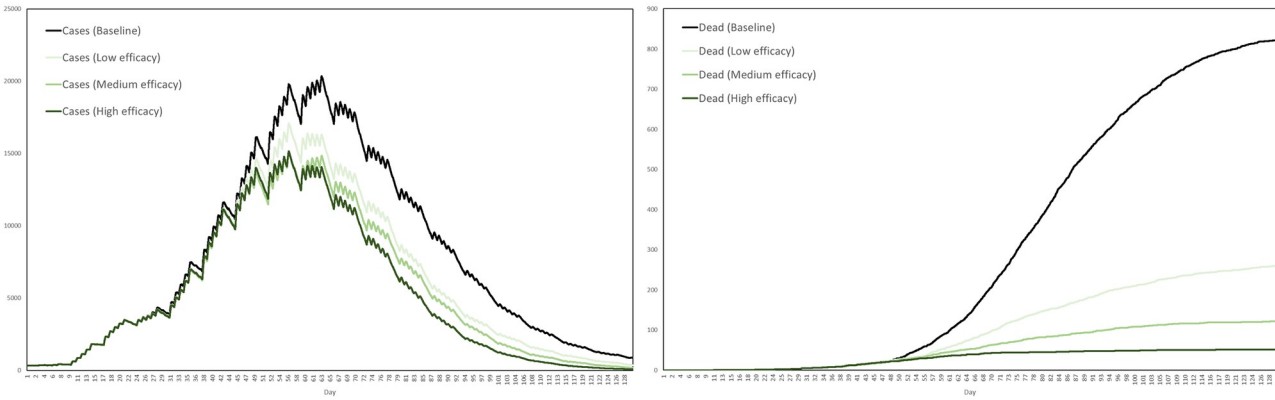

**Fig 29. Impact of vaccine efficacy on cases and deaths.** Low, medium and high efficacies correspond to probabilities of protection against infection after two doses of 0.55, 0.75 and 0.95, respectively.

hospital workers, followed by all other agents according to age, down to a minimum age of 16. We will assume no vaccine hesitancy and that the number of first doses available each day is equivalent to 0.6% of the total population. In the Luxembourg implementation, this yields a constant daily capacity of 4864 first doses. We will assume that each vaccine is administered in two doses, precisely 3 weeks apart. We will consider three vaccines, of low, medium and high efficacy, for which we assume that after the first dose these vaccines provide protection against infection with probabilities 0.450, 0.675 and 0.900, respectively, increasing after the second dose to 0.55, 0.75 and 0.95, respectively. If $p_1$ and $p_2$ denote the probabilities that the first and second doses successfully protect against infection, then the values of the pair $(p_1, p_2)$ corresponding to the low, medium and high efficacies are therefore (0.450, 0.182), (0.675, 0.231) and (0.900, 0.500) since, according to our simple model of vaccination, if administered as a single dose the vaccines have efficacy $p_1$ while after two doses the efficacy increases to $p_1 + (1 - p_1)p_2$. For each of the three vaccines we performed ten simulations and averaged the resulting numbers of cases and deaths, plotting the results in Fig 29.

We see from Fig 29 that, vaccinating in the midst of an outbreak, the impact on cases is small, but the impact on deaths is high, even for the low efficacy vaccine.

**Capacity.** We now look at the impact of lower and higher daily capacity. We take the medium efficacy vaccine, administer it according to the same strategy and assume no vaccine hesitancy. We set low, medium and high daily first dose availability equivalent to 0.2%, 0.6% and 1.0% of the total population, respectively, resulting in the Luxembourg implementation at daily first dose capacities of 1621, 4864 and 8107, respectively. Performing ten simulations for each scenario, we average cases and deaths and plot the results in Fig 30.

We see from Fig 30 that even a low daily first dose capacity has a significant impact on reducing deaths. As with efficacy, we see that the impact on cases is relatively small in comparison to the impact on deaths.

**Hesitancy.** For the medium efficacy vaccine with medium daily capacity, administered according to the same strategy, we now consider the impact of low, medium and high levels of vaccine hesitancy. In particular, we assume that with a certain probability agents refuse the vaccine when offered it. We assume that these probabilities are age dependent and that they remain constant throughout the simulation. An online survey conducted by science.lu in Luxembourg in December 2021 [97] suggested that vaccine hesitancy levels were fairly high in Luxembourg, with only 55% of participants being likely or very likely to get a COVID-19 vaccine. Breaking down by age, the survey suggested that in the age group 13-34, only 48% were

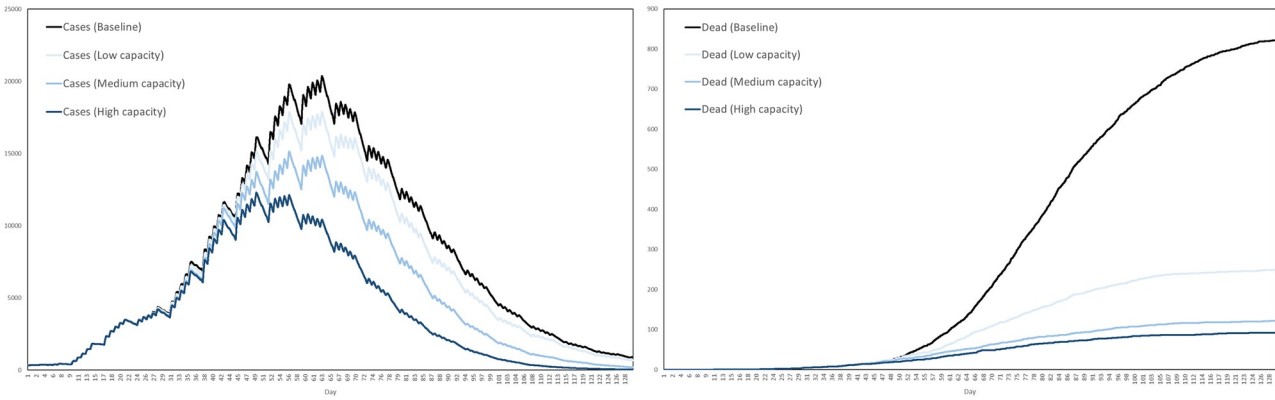

**Fig 30. Impact of daily first dose capacity on cases and deaths.** Low, medium and high capacities correspond to daily first dose availabilities equivalent to 0.2%, 0.6% and 1.0% of the total population, respectively.

likely or very likely to get vaccinated, 57% in the age group 35-64 and 80% in the age group 65+.

For our simulations, we decompose according to the same age groups with low, medium and high vaccine hesitancy levels for each age group parametrized as in Table 12. For example, for the low hesitancy scenario we assume that agents aged 65+ refuse the vaccine with probability 0.10, while for the high hesitancy scenario we assume agents aged 16-34 refuse the vaccine with probability 0.75, representing the two extremes. The medium hesitancy scenario corresponds roughly to the data collected in the Luxembourg survey, while the probabilities for the low and high hesitancy scenarios are obtained by interpolating half way between the medium scenario and the two extreme cases of zero and total hesitancy.

Performing 10 simulations for each of the three scenarios, we plot the average numbers of cases and dead in Fig 31 together with those of the baseline.

We see from Fig 31 that high levels of hesitancy result in considerably more deaths than medium or low hesitancy. That being said, the levels of hesitancy corresponding to our high hesitancy scenario are, in some sense, very high. We assumed hesitancy levels to be constant throughout the simulation, although in reality hesitancy levels can change over time. For example, as more people are vaccinated, hesitancy levels might decrease as familiarity with the vaccine increases. On the other hand, as more people are vaccinated the likelihood of somebody experiencing unusual side effects of the vaccine increases, with news of this potentially increasing hesitancy levels. While we have assumed a model of vaccine hesitancy that acts on the level of the individual, hesitancy can also manifest itself at a higher level, with policy makers themselves hesitant to implement the vaccine. Moreover, we have only simulated the use of a single vaccine. A future study would have several being administered simultaneously, starting on different dates, with different properties and with potentially different levels of hesitancy associated to them. Such considerations are beyond the scope of the present study.

**Table 12. Probabilities of vaccine refusal in low, medium and high hesitancy scenarios.**

|  |  | Hesitancy | | |
|---|---|---|---|---|
|  |  | **Low** | **Medium** | **High** |
| **Age** | 16-34 | 0.25 | 0.50 | 0.75 |
|  | 35-64 | 0.20 | 0.40 | 0.70 |
|  | 65+ | 0.10 | 0.20 | 0.60 |

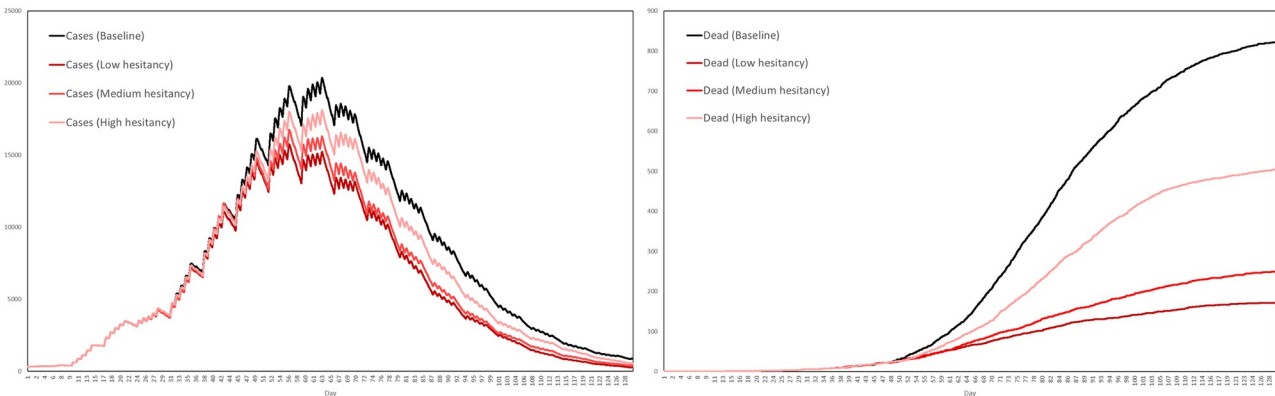

**Fig 31. Impact of vaccine hesitancy on cases and deaths.** Low, medium and high hesitancy levels are assumed to be age-dependent.

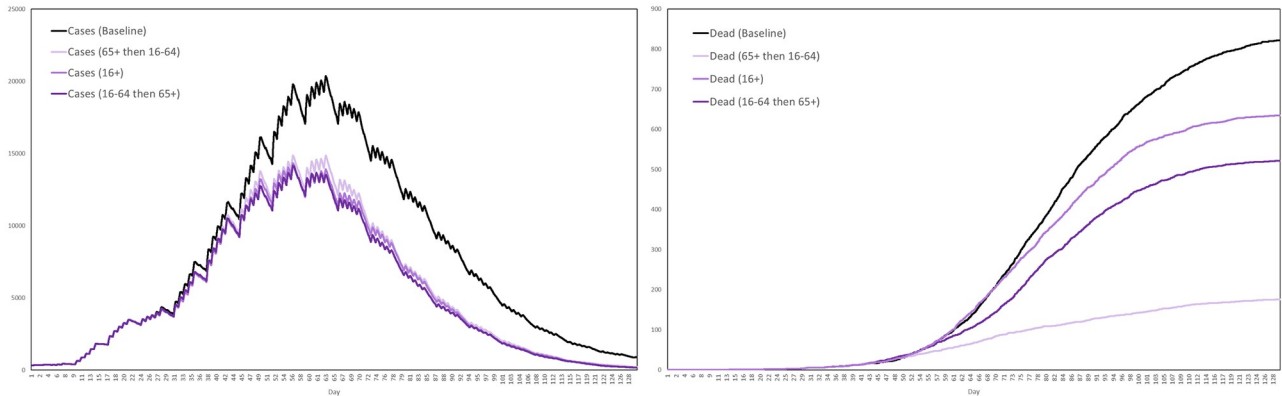

**Fig 32. Impact of vaccine strategy on cases and deaths.** We consider the three cases in which vaccines are distributed to old then young, young and old together, and finally young then old.

**Strategy.** Finally, for the medium efficacy vaccine with medium daily capacity and no hesitancy, we now consider three different allocation strategies. The first, a simplified version of the priority scheme used in the other experiments, first allocates vaccines to the age group 65+ and then to the age group 16-64, proceeding in a random order within each group. The second distributes vaccines randomly to the entire age group 16+. The third starts with 16-64 and then moves onto 65+, the opposite of the first strategy. We expected that the strategy that prioritizes young people would lead to the largest reduction in cases, while the strategy that prioritizes old people would lead to the largest reduction in deaths. For each scenario, we performed 10 simulations and plot the average numbers of cases and dead in Fig 32, comparing to the baseline.

Fig 32 suggests that vaccinating younger people in an attempt to reduce transmission and therefore deaths is not as effective as simply vaccinating the elderly first, since it leads to a much smaller reduction in deaths while resulting in only a very minor improvement in case numbers.

## Conclusion

Based on the results presented and discussed in the previous section, we now draw several conclusions. We do so keeping in mind the limitations of our model, and the assumptions on which it is based. Our basic conclusions are listed as follows:

- Our agent-based model predicts far fewer cases than the basic SEIR model. The latter assumes homogeneous mixing and therefore represents only an upper bound, with the heterogeneities captured by our model explaining the difference. Our model predicts only around 25% as many cases as the SEIR model.

- Testing and contract tracing flatten the epidemic peak, but their impact in reducing deaths is limited.

- A full lockdown, although economically and socially very costly, dramatically reduces both cases and deaths. Alternatives to the full lockdown are also available, not as effective but less costly in terms of their economic disruption. The impact of an 11pm-6am curfew is relatively small.

- When vaccinating against a future outbreak, herd immunity is achieved at levels much lower than those predicted by the simple SEIR model. Under certain assumptions, our model predicts that substantial levels of protection are achieved with only 30% of the population immune.

- When vaccinating in midst of an outbreak, the task is more difficult. In this context, the impact of vaccination in reducing cases is less, however the impact in reducing deaths remains high. A low efficacy vaccine is almost as good as a high efficacy vaccine, from the point of view of reducing deaths, although this conclusion is somewhat dependent on our interpretation of efficacy. As regards daily capacity, even with only a low number of doses administered each day the impact on deaths can be relatively high, so long as these doses are targeted at the most vulnerable individuals. High vaccine hesitancy results in considerably more deaths than would occur with low vaccine hesitancy and is the most serious challenge to a successful vaccination programme.

While in the previous section we considered independent variations in vaccine efficacy, daily capacity and hesitancy, in order to assess their individual impact, it is also worth considering the impact of a mixed variation of these parameters. In particular, we consider also the best and worse case scenarios, with the best case corresponding to high efficacy, high capacity and low hesitancy and the worse case corresponding to low efficacy, low capacity and high hesitancy. Performing ten simulations for each scenario, starting the vaccinations 3 weeks into the outbreak as before, we plot the average cases and deaths in Fig 33, as well as the averages for the baseline scenario in which no vaccination occurs. What we conclude from this is that in the worst case scenario the vaccination programme essentially fails, while in the best case scenario the vaccination programme is extremely successful at reducing deaths, the main factor here being vaccine hesitancy, with efficacy and capacity being nonetheless significant. Even in the best case scenario, when vaccinating in the midst of an explosive outbreak, there will still be large numbers of new cases many weeks after the start of the vaccination programme, however the peak will be smaller and occur sooner.

Let us finish with some final remarks about the limitations of the model and directions for future research. Firstly, confidence in our results would be further improved with a more sophisticated statistical analysis of uncertainty, which would require several computational optimizations, in order to reduce runtime. This may involve validating our model using data collected in other countries, besides only Luxembourg. Obtaining all the data necessary to do this is difficult, and was therefore deemed beyond the scope of the present work.

Moreover our behavioural model could be further enhanced, by introducing correlations between individuals, for example families, and with agent behaviour adjusting automatically in response to a pandemic event, even in the absence of organized interventions.

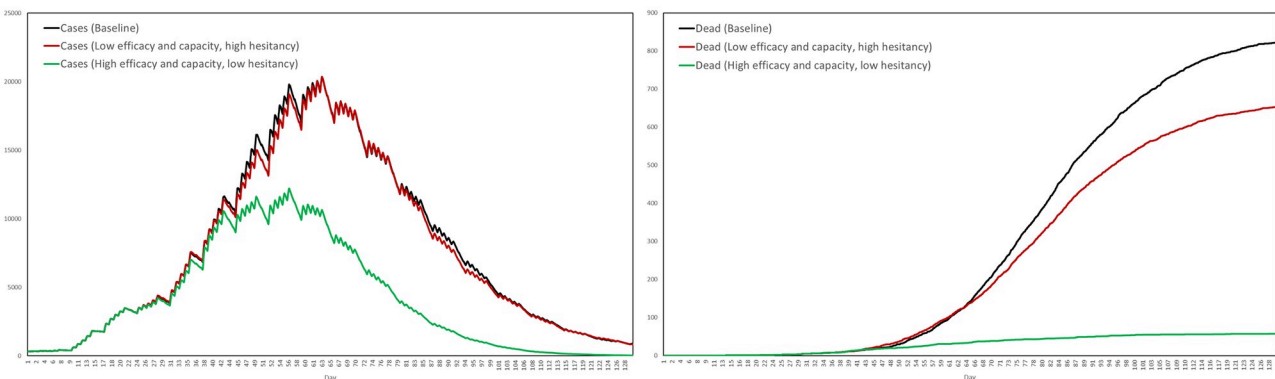

**Fig 33. Best and worst case vaccination scenarios.** The best case corresponds to high efficacy, high capacity and low hesitancy. The worse case corresponds to low efficacy, low capacity and high hesitancy. For each scenario we performed 10 simulations and plotted the mean numbers of cases and deaths, together with the baseline.

One notable limitation of our study, is that our interpretation of vaccine efficacy is relatively simple, based on an all-or-nothing model of immunity. A more sophisticated model of immunity, distinguishing immunity against infection from immunity against transmission, and also immunity against symptomatic illness or severe disease, would be more realistic and could lead to interesting conclusions. We have also not considered factors such as partial immunity, waning immunity, cross immunity for multiple variants and multiple vaccines. The all-or-nothing model of immunity is assumed for simplicity, with this assumption often being found in the literature.

Furthermore, new strains of COVID-19 present new challenges, but we have not simulated the impact of different strains, nor attempted to model competition between strains. We speculate that social distancing and testing exert an evolutionary pressure on the virus that increases the reward for any mutation that makes the virus more transmissible or less easily detected. The simulation of such a competitive system is an objective for future research, with uncertainties surrounding the strains a major reason why we have not made any concrete predictions about the future. Since we are ourselves a part of the system that we are trying to model, and therefore not independent from it, to a certain extent we would be doomed to fail anyway.

Nonetheless, our results reinforce the widely held view that vaccination is the most effective intervention against COVID-19. Lockdowns are extremely costly, both socially and economically, with other non-pharmaceutical interventions having only a limited impact. Vaccination represents the best hope we have to free ourselves from this deadly virus, our final conclusion being that a positive and progressive approach to vaccination is essential.

## Appendix

In this appendix we describe our model according to the ODD protocol. The generic parametrization of submodels is described in the Methods section, while parametrizations specific to particular scenarios are described in the model evaluation and results sections, so we will not repeat those details here.

### Purpose and patterns

The purpose of this model is to explore the impact of interventions, in particular vaccination, on cases and deaths due to COVID-19. The intention is to help decision makers understand

the relative strengths of interventions when used in combination with one another. The model has been configured to represent Luxembourg and therefore the patterns that the model has been assessed against were observed in Luxembourg during the first few months of the pandemic. This includes, in particular, the drops in cases and deaths seen after multiple strict measures were introduced in March 2020.

## Entities, state variables and scales

The basic entities in our model are agents and locations:

- **Agents**: The agents represent individuals living or working in a given region. They are assigned age, health state, nationality and lists of locations at which they are able to perform various activities. In addition to these state variables, agents are assigned a behavioural routine describing which activities they perform and when they perform them, the time resolution being 10 minutes.

- **Locations**: The locations represent places where the agents can perform activities. Locations are assigned spatial coordinates and a type, with the possible types of location listed in Tables 1 and 2. Coordinates are assigned by sampling population grid data. The grid data has a resolution of 1 kilometre, with the coordinates sampled from this in WGS84 format at a resolution of 1 meter.

## Process overview and scheduling

Our model is configured to run for a fixed number of iterations, with each iteration representing a 10 minute interval of time. During each iteration of this main loop, interventions are updated according to a schedule and internal message and telemetry buses are notified of world updates occurring since the last tick of the simulation clock. Components are notified of the new time, to which they might then respond. For example, it might be time for the movement model to request that an agent moves to visit a care home, but a lockdown intervention listening to such requests overrides the request, requesting instead that the agent returns home. The disease model loops over all locations and determines if any new infections take place, requesting health state updates if so. Once these requests have been resolved via the message bus, world updates are enacted and the simulation moves onto the next tick, with the simulation finally ending after the predetermined number of ticks.

## Design concepts

**Basic principles.** The model implements a conventional compartmental disease model within the bottom-up approach of an agent-based model. The compartmental disease model is familiar and easily understood, while the agent-based approach provides a more detailed and flexible model of social interactions than can be achieved with the equation-based approach. In particular, the agent-based approach allows for an intuitive and realistic implementation of interventions. This is much more difficult to achieve at the aggregated level of a small system of differential equations. Another basic principle of our model, and one that influenced its design, is adaptability. Our model is built on a modular framework, with components communicating with one another via a message bus, having the advantage that components can be easily added or replaced, transforming the model with ease to describe new regions, diseases or interventions.

**Emergence.** Emergence is a concept that sits at the heart of our approach to modelling. Behaviour is described on an individual basis, with routines sampled from a pool of over 2000

possibilities, yielding an extremely complex system of collation and movement. By simulating an infectious disease spreading within such as system, we observe the resulting epidemic as an emergent phenomenon. The set of all possible sequences of interactions between agents is extremely large, with certain sequences having a dramatic effect on the total number of deaths. A chain of interactions ending in a care home might, for example, be of this type.

**Adaptation.** Agents in our model do not adapt their routines willingly. If a routine is disrupted, it is because an intervention has over-ridden it. In other words, in the absence of interventions agents will behave as if oblivious to the epidemic. Adaptive routines, based on learning objectives and prediction might enhance the model, but would be very difficult to parametrize.

**Sensing.** Components, such as the disease model and the interventions, collect data on the world and respond accordingly. This is achieved via the message bus, the system of information exchange to which components can subscribe and publish events. The stream of communications between the components results from the interactions of the agents and the disease model, and therefore represents an emergent collection of events.

**Interaction.** If two agents occupy the same location for the same 10 minute interval, then it is assumed that an interaction occurs that with some probability results in disease transmission. The nature of this interaction is assumed to be uniform across all location types. While in reality location type or activity might be important factors in determining the probability of transmission, in the absence of relevant data we make no such hypotheses, assuming uniformity of interactions for simplicity.

**Stochasticity.** Stochasticity is used throughout our model, during both initialization and simulation. The world is procedurally generated, with locations distributed and populated by sampling probability distributions. For each agent, movement is determined by the random selection of locations belonging to certain lists, while disease transmission is also the result of random, binomial, sampling. Via repeated sampling, stochasticity washes away outliers that may arise form a particular configuration. Much care was taken to ensure that our experiments can be repeated and the results replicated, by keeping track of the random seeds used by the psuedo-random number generators appearing in our code.

**Collectives.** Agents routines are sampled from a finite pool, and therefore some agents behave very similarly. In addition, agents living in the same house will tend to visit similar nearby locations. These correlations, however, are not the result of emergent collective behaviour, being instead consequences of the configuration process.

**Observation.** A telemetry system observes and collects data on each simulation. The systems consists of reporters, each of which looks at a different aspect of the simulation. The reporters are as follows:

- **Health State Counts**: This reporter records, at each tick, the numbers of resident agents in each health state.

- **Activity Counts**: This reporter records, at each tick, the numbers of resident agents performing each activity.

- **Location Type Counts**: This reporter records, at each tick, the numbers of resident agents in each type of location.

- **Testing Counts**: This reporter records, at the end of each day, how many tests and positive tests were performed that day, distinguishing between residents and non-residents.

- **Testing Events**: Each time a test occurs, this reporter records the date and time, the test result, the agent's age and health state, the residency status of the agent and the coordinates of their home.

- **Quarantine Counts**: This reporter records, at the end of each day, how many agents are in quarantine. It also calculates the average age of these agents and breaks them down by health state.

- **Exposure Events**: Each time a new infection occurs, this reporter records the date and time, the type of location and who infected who. It records the ages of the two agents and which activities they were each performing at the time.

- **Death Events**: Each time a agent dies, this reporter records the date and time, their age, whether they live in a house or a care home, and information on their place of work.

- **Vaccination Events**: Each time a first dose of a vaccine is administered, this reporter records information about the agent in question, including age, health state and household composition.

- **Secondary Infection Counts**: Throughout the simulation, this reporter counts how many infections each agents causes. At the end of this simulation, it then calculates a histogram, illustrating the distribution of secondary infection counts, from which a mean can then be derived.

## Initialization

Initialization begins by creating a map of the region. This includes a model of population density. This is followed by the creation of the world, based on the map, which involves distributing locations and populating them with agents. Having created the world and a clock object, to keep track of time, the remaining components of the model are then initialized. These components are the disease model, the activity model, the movement model and the interventions. For example, during the initialization phase it is determined who will die if infected, who will work night shifts and who will refuse a vaccine. With the initialization phase completed, the simulation begins.

   More precisely, having constructed the map object, the world is built in the following order:

- Resident agents are created and assigned an age and nationality.

- Locations are created and assigned coordinates.

- Resident agents are assigned homes, with the most elderly being assigned care homes. The mechanism by which agents are grouped into households reflects an expected distribution of ages derived from census data.

- Neighbouring countries and populations of cross-border workers are created, with these adults being assigned an age and nationality. These agents will perform all activities other than work in their home country.

- Agents are assigned a place of work, to which they will move if performing the work activity.

- Resident agents are assigned a number of homes, shops and restaurants that they may visit during the simulation. These are sampled in terms of the distance to the agent's home.

- Resident agents are assigned a number of cinemas or theatres and museums or zoos that they may visit during the simulation. These are sampled randomly from all such locations in the region.

- Resident agents are assigned primary and secondary schools, to which they move if performing the school activity, and also a medical clinic, place of worship and indoor sports center.

Locations of these types are assigned based on proximity, unless the location has already been assigned its fair share of agents, in which case the next nearest available location is chosen. This is to avoid overcrowding, ensuring that the number of agents visiting these locations is balanced.

- Resident agents are assigned cars, with households being given one car each.

The procedure described above therefore assigns to each agent and for each activity a list of locations from which the agent can randomly choose when performing that activity during the simulation. It therefore remains to initialize the aforementioned components:

- The disease model assigns to each agent a disease profile, describing the trajectory of health states through which the agent will pass should they be infected, and an associated list of durations, indicating how long the agent will spend in those states. A number of resident agents are randomly infected and their health state set accordingly. These will be the initial cases that get the epidemic started.

- The activity model assigns to each agent a weekly routine, sampled from over 2000 such routines with a 10 minute resolution. These routines distinguish between weekdays and weekends. The initial activity of each agent is set accordingly, together with an initial location.

- The contact tracing system initializes, determining for each agent a list of regular contacts. This is a list of other agents who live, work or go to school with the given agent. These contacts will be subject to quarantine and testing should the agent test positive during the simulation.

- The test laboratory, test booking and prescription testing systems initialize, collecting information on health states from the disease model. The large scale testing intervention assigns to each agent a period of time that the agent will wait before responding to a test invitation, should such an invitation be received.

- The location closure interventions initialize. In the case of care homes, this involves creating lists of agents working in each care home.

- The vaccination intervention constructs an ordered list of agents to be vaccinated during the simulation. During this initialization phase, it is determined which agents will refuse vaccination and therefore be omitted from the list.

- The curfew and hospitalization interventions initialize, although do not require any detailed procedures.

## Input data

The model uses several sources of input data. Some are used to configure time varying processes. The activity routines, assigned during the initialization phase, describe the sequence of activities performed by each agent, constructed from time use data obtained by STATEC [16]. The numbers of trains, buses and trams operating through the day is variable and configured within the movement model, using data obtained by Mobilitéit [17]. Moreover, each intervention operates according to a schedule, consisting of dates on which to enable or disable the intervention or on which to update the values of certain parameters. This uses COVID-19 surveillance data, derived from a national database managed by the General Inspectorate of Social Security in Luxembourg.

## Submodels

The model includes a number of submodels, the most important of which are listed as follows (some of which are described in more detail in the Methods section):

- **Map Factory**: The map factory compiles population grid data to produce a distribution from which location coordinates can be sampled. It includes a subsystem that refines this distribution via linear interpolation, improving the resolution beyond the default 1 kilometre.

- **World Factory**: The world factory creates agents and locations and for each agent assigns for each activity a list of locations to which the agent can move during the simulation. These lists are determined beforehand since otherwise the computational cost would be too high when dealing with large populations.

- **Message Bus**: The message bus allows components to communicate through a shared set of interfaces. Communications are either requests or notifications. Requests are made to, for example, begin a new activity, move to a new location or book a test. Other components might cancel these requests, issuing their own requests in response. Once such disagreements are all resolved, with the state of the world updated accordingly, notifications are sent through the message bus informing components of these changes. The message bus was implemented to account for the fact that interventions must interact with one another when several are simultaneously active. There is also a telemetry system, operating on the same principles as the message bus, that collects and saves data from the simulation for analysis.

- **Clock**: The clock keeps track of the time, both in terms of ticks and in ISO 8601 format. In the default configuration, a tick of the clock represents an interval of length 10 minutes. Components keep track of the current time via the message bus.

- **Deferred Event Pool**: This object stores events due to occur at a later time in the simulation. For example, once an agent has received their first dose of the vaccine, the administration of the second dose is added to the deferred event pool, as an event due to take place on a particular date several weeks after the first. On that date, the system will then issue a request to the message bus, triggering the vaccination system to actually perform the second dose.

- **Scheduler**: The scheduler is a system that parses input data on dates and parameter values to produce for each intervention an implementation that varies over time. This is necessary since model validation requires the reproduction of measures introduced in Luxembourg during the first months of the COVID-19 pandemic, with various quantities associated to these measures being variable. For example, daily testing was variable, while places of work, schools and other locations were closed on certain dates and reopened on others.

- **Disease Model**: The disease model was designed according to the familiar compartmental framework but in such as way that avoids geometrically distributed periods of time spent in each health state. Rather than using stochasticity on each tick to decide who moves into the next health state, disease progression for each agent is determined during initialization, allowing for a richer and more realistic variety of patterns. On each tick, the transmission model loops through all locations and determines who, if anyone, is to be newly exposed. More precisely, it counts how many infectious agents are in a given location, distinguishing between symptomatics and asymptomatics, and loops through the susceptible agents in that location, sampling binomial distributions to determine if those agents are to be infected. If infections occur, the system then decides, via random selection, who exactly caused each infection. The algorithm is so ordered to optimize runtime, with the identification of the infecting agent needed only for telemetry and testing purposes.

- **Activity Model**: The activity model was designed to give agents interesting, varied and realistic daily and weekly routines. Assigning these routines during initialization lowers the computational cost, versus a system that for each agent chooses activities stochastically. Such a system, based on Markov chains, was previously implemented in our model, but was replaced due to the computational burden and the fact that, after repeated testing, did not appear to be sufficiently advantageous.

- **Movement Model**: As stated above, the world factory determines lists of locations that agents might visit. In the event that that an agent starts a new activity, the movement model simply selects a location at random from the appropriate list.

- **Hospitalization**: This hospitalization intervention moves agents to hospital if their health transitions to a state demanding hospitalization. This intervention is relatively simple, and does feature hospital or ICU capacity, such constraints being omitted due to parametrization uncertainties. The hospitalization intervention also takes care of agents who have died, moving them to the cemetery. Dead agents are moved to the cemetery to avoid them being erroneously counted as inhabiting other locations.

- **Test Booking System**: The testing system is quite large and therefore divided into several subsystems. The test booking system handles requests to get tested. The test events themselves are scheduled via the deferred event pool.

- **Testing Laboratory**: The laboratory system performs the tests, handling deferred test event requests, published through the message bus. If the daily limit of tests has been reached, then subsequent tests that day are simply not performed. If a test takes place, the result of the test is published to the message bus for other components to see.

- **Prescription Testing**: Tests are booked in our model for one of two reasons. The first is that an agent has developed symptoms, detected if a health state transition has been published to the message bus in which an agent is symptomatic having not previously been so.

- **Large Scale Testing**: The other circumstance in which an agent books a test is after they have been invited to do so by the large scale testing system. For simplicity, our implementation of this system distributed tests at random. Once an invitation has been received, agents respond by booking a test after a delay. It was important to include this delay since data collected in Luxembourg shows that this period of time is often quite substantial.

- **Contact Tracing**: The contact tracing system responds to newly published test results. If the result of a test is positive, the system issues test booking and quarantine requests to regular contacts of the relevant agent. More detailed implementations of contact tracing are possible, and were tested, however the system described seems to provide a good balance between realism and runtime when simulating very large numbers of individuals.

- **Quarantine**: The quarantine model holds a list of agents who are subject to quarantine restrictions. Agents are added to the list if a quarantine request is made, which occurs either via the contact tracing system or if an agent tests positive. Agents are removed form the list once their period of quarantine is over, a period which can be reduced if the agent should happen to get a negative test result. The quarantine system interacts with the movement model by overriding requests to leave home if an agent is in the quarantine list. In particular, we assume that agents completely adhere to the quarantine rules.

- **Location Closure**: The location closure system interacts with the movement model in a way that is similar to the quarantine system. If an agent requests to move to a location that is, as

determined by the scheduler, currently off limits, then that request is denied with the agent being sent home instead. The only exception here is care homes, with agents still permitted access to a care home if they happen to work there.

- **Curfew**: The curfew system is very similar to the location closure system, acting on list of disallowed locations which in this case includes everything except hospitals and the cemetery. The difference is that the curfew, on days when it is enabled, is only active between certain hours.

- **Vaccination**: This system incorporates several features that were deemed to be of most importance. One such feature is vaccine hesitancy, representing the fact that not everybody wants to get vaccinated. The probability of refusal is determined by age. Another is variable efficacy, representing an increased efficacy after the second dose of a two-dose vaccine. The system also features a priority list, representing systems in which limited supplies of vaccine are allocated to certain individuals before others, according to age, residency or place of work. Vaccinations run on a daily cycle, with a deferred event pool and the message bus being used to schedule the administration of second doses. The vaccination model was designed to encompass such a level of detail since an examination of the impact of vaccination was one of the main objectives of the study.

## Acknowledgments

The authors would like to thank Dr. Mikołaj J. Kasprzak for his help in drafting the grant proposal, communications with STATEC, preparing the timeline used for Fig 13 and for pointing the authors towards several useful references. The authors would also like to thank Dr. Christian Selinger for several useful discussions and for drawing our attention to the generation time as a quantity of interest.

## Author Contributions

**Conceptualization:** James Thompson.

**Data curation:** James Thompson, Stephen Wattam.

**Formal analysis:** James Thompson.

**Funding acquisition:** James Thompson.

**Investigation:** James Thompson, Stephen Wattam.

**Methodology:** James Thompson, Stephen Wattam.

**Project administration:** James Thompson.

**Resources:** Stephen Wattam.

**Software:** James Thompson, Stephen Wattam.

**Supervision:** James Thompson.

**Validation:** James Thompson, Stephen Wattam.

**Visualization:** James Thompson, Stephen Wattam.

**Writing – original draft:** James Thompson.

**Writing – review & editing:** James Thompson, Stephen Wattam.

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
