## [Decision Letter · Decision Letter 0]

5 May 2021

PONE-D-21-09294

Estimating the impact of interventions against COVID-19: from lockdown to vaccination

PLOS ONE

Dear Dr. Thompson,

Thank you for submitting your manuscript to PLOS ONE. After careful consideration, we feel that it has merit but does not fully meet PLOS ONE’s publication criteria as it currently stands. Therefore, we invite you to submit a revised version of the manuscript that addresses the points raised during the review process.

Based on the reviewers' comments, I recommend that you make substantial revisions to your manuscript. This should include improving the description of the model environment and how the agents interact in it. Please also be clear on how the sensitivity analysis was carried out and in reporting the outcome of this analysis. Then resubmit your improved version of the manuscript after making the revisions as advised in this letter.

We look forward to receiving your revised manuscript.

Kind regards,

Martin Chtolongo Simuunza, PhD

Academic Editor

PLOS ONE

Journal Requirements:

2. In your Methods section, please ensure that sufficient information to make the study reproducible are provided (for example, by reporting the equations  representing the model, and describing parameters and assumptions applied).

5. Thank you for stating the following in the Financial Disclosure section:

[This project was funded by a grant from the COVID-19 Fast-Track program of the Fonds National de la Recherche Luxembourg (www.fnr.lu). The grant number is COVID-19/2020-2/14858807/ABMLUX, with the grant being awarded to Dr. J. Thompson. The funders had no role in study design, data collection and analysis, decision to publish, or preparation of the manuscript.].   

We note that one or more of the authors are employed by a commercial company: W&P Academic Consultancy

6. We note you have included a table to which you do not refer in the text of your manuscript. Please ensure that you refer to Table 5 in your text; if accepted, production will need this reference to link the reader to the Table.

7. We note that Figure 2 in your submission contain map images which may be copyrighted. All PLOS content is published under the Creative Commons Attribution License (CC BY 4.0), which means that the manuscript, images, and Supporting Information files will be freely available online, and any third party is permitted to access, download, copy, distribute, and use these materials in any way, even commercially, with proper attribution. For these reasons, we cannot publish previously copyrighted maps or satellite images created using proprietary data, such as Google software (Google Maps, Street View, and Earth). For more information, see our copyright guidelines: http://journals.plos.org/plosone/s/licenses-and-copyright.

You may seek permission from the original copyright holder of Figure 2 to publish the content specifically under the CC BY 4.0 license. 

If you are unable to obtain permission from the original copyright holder to publish these figures under the CC BY 4.0 license or if the copyright holder’s requirements are incompatible with the CC BY 4.0 license, please either i) remove the figure or ii) supply a replacement figure that complies with the CC BY 4.0 license. Please check copyright information on all replacement figures and update the figure caption with source information. If applicable, please specify in the figure caption text when a figure is similar but not identical to the original image and is therefore for illustrative purposes only.

Reviewers' comments:

Reviewer's Responses to Questions

**Comments to the Author**

1. Is the manuscript technically sound, and do the data support the conclusions?

Reviewer #1: Yes

Reviewer #2: Yes

2. Has the statistical analysis been performed appropriately and rigorously? 

Reviewer #1: No

Reviewer #2: No

3. Have the authors made all data underlying the findings in their manuscript fully available?

Reviewer #1: Yes

Reviewer #2: No

4. Is the manuscript presented in an intelligible fashion and written in standard English?

Reviewer #1: Yes

Reviewer #2: Yes

5. Review Comments to the Author

Reviewer #1: The author presented a sophisticated, detailed agent-based model of Covid-19 transmission in Luxembourg. It evaluated a number of interventions to control the spread of Covid-19 in the community and compared the outcomes with other published models in the discussions. The study covered an important topic and included the verification and validation which help build confidence in the model. The inclusion of ODD protocol for the model help reading and understanding the model descriptions easier and more efficient and make it easier to replicate.

There are a number of issues that I would like the authors to clarify:

1. The paper discussed the sensitivity of the number of dead to the distribution of care homes, and the sensitivity of our model with respect to the probability of being asymptomatic and the transmission probability.

Which parameters is each model outcome sensitive to?

Which approach did the authors use to perform sensitivity analysis (e.g. univariate, multivariate, or probabilistic)?

I suggest the authors to include a table that summarizes the values of model parameters in the baseline scenario and the distributions used in sensitivity analysis.

2. How was the vaccine efficacy implemented in the model not very clear?

- Did the authors refer to the vaccine efficacy in this paper as efficacy against confirmed Covid-19 case or efficacy against transmission?

- What state variables of agents were affected if they are vaccinated? (Did the vaccinated agents have their state of infection changed from Susceptible to Recovered/Immune at the probability p1 and p2?)

- How did vaccination affect the transmission if susceptible vaccinated individuals come into contact with infected individuals and their disease progression if infected?

3. The number of simulations (ten) performed for each scenario seem quite small for a highly stochastic model like this model. I appreciate the extensive runtime for this high-resolution agent-based model. However, how did you decide that 10 simulations were sufficient to capture the true behaviours of the system? Would reducing the resolution and increasing the number of simulations produce outcomes with less uncertainty?

4. I appreciate that it is a very sophisticated and detailed model. However, would the purposes of the study be achieved with a simpler model such as a stochastic compartment model or an ABM with simpler structure? The authors mentioned that some of their results were similar to results from other studies which using simpler models. In this case, what would be the justification for building a more complex model?

Reviewer #2: The paper presents a detailed agent-based model that simulates COVID-19 pandemic dynamics over a small country, drawing very interesting results. The proposed system has agents with a high level of detail of the human interactions and routines, a great diversity of spatial locations and their intrinsic impacts, and it is very customizable. The methodology includes a consistent estimation of the parameters and used the classic SEIR model as a baseline, showing higher accurate inferential performance. However, some improvements must be made before its publishing.

The overall text is well written but sometimes exaggerates in length, and many paragraphs could be replaced by a diagram, a table, etc, reducing the length of the text and yet improving the readability of the results.

There is a problem with the location and quality of the images in the review text that harmed the reading and understanding of the text. All images appeared at the end of the text, which is inopportune. I suggest fixing this issue for the next review rounds.

In the State of Art section I think is missing a lot of recent machine learning and ABM models. This section can be improved with a deeper revision of recent research of agent-based epidemiological simulation of COVD19.

In the Methods section, the authors focused on the semantic meaning of the system components, which is perfectly understandable given the scope of the research. However, a little bit more details about the software implementation could improve the text and help the readers interested in implement or even employ the model. On the other hand, some descriptions of the model behavior (e. g. the school and classes descriptions) could be abbreviated.

Or you simply share the link of the source code or you should remove from the text the promise of opening the code in the future.

The authors let it clear in the text that the ABM has a network structure. But the Locations subsection gives the impression that the software is grid-oriented. How explicitly occur the relationships between the locations and the agents? How are they represented? Along with the text, the authors recovered the notion of a network model but in a fashion hard to understand. A good example is a car using by the agents. There will be a unique car location shared by all agents? I think that including a Network Diagram in this section, showing a small sample of the agents and their interactions will help the readers to understand better how your model works.

The ODD protocol in Appendix has an interesting high-level description of the whole system. However, I still miss some logical diagrams of the software structure and architecture (maybe a UML package diagram, or a component diagram ?)

In Model Evaluation and Validation, much of these paragraphs can be simplified or synthesized using mathematical formulas or replaced by a table with the parameters and their sources.

The results are very interesting but are spread in text and should be synthesized in a table, with the results of the proposed model and the SEIR model side by side, together with a statistical significance test of the difference between them.

Finally, I think this research is very interesting and opportune and their insights are very important and worth publishing after improving the paper.

6. PLOS authors have the option to publish the peer review history of their article (what does this mean?). If published, this will include your full peer review and any attached files.

Reviewer #1: No

Reviewer #2: No

---

## [Author Response · Author response to Decision Letter 0]

13 Oct 2021

My co-author and I welcome the comments on our paper `Estimating the impact of interventions against COVID-19: from lockdown to vaccination'. We appreciate the thorough and insightful quality of the reviews and have improved our article accordingly, with substantial revisions. We have improved the description of the model environment and how the agents interact in it, and released the code for our model in a public repository:

https://github.com/abm-covid-lux/abmlux

All input and output data, underlying the findings in the manuscript, are now fully available. The input data can be found in the above repository, while the output data, used to create the plots and tables in the manuscript, is located in a separate public repository:

https://github.com/abm-covid-lux/output.

We have now made clear how the sensitivity analysis was carried out and reported the outcome of this analysis. We have performed 790 additional simulations, on top of the 290 performed for the original manuscript, and included additional statistical analysis.

Detailed responses to the comments of the reviewers are as follows:

Reviewer #1

1. `The paper discussed the sensitivity of the number of dead to the distribution of care homes, and the sensitivity of our model with respect to the probability of being asymptomatic and the transmission probability. Which parameters is each model outcome sensitive to? Which approach did the authors use to perform sensitivity analysis (e.g. univariate, multivariate, or probabilistic)? I suggest the authors to include a table that summarizes the values of model parameters in the baseline scenario and the distributions used in sensitivity analysis.'

Response: We have added a subsection on Sensitivity Analysis to end of the Model Evaluation section. The analysis performed was univariate. We estimated the partial derivatives of total deaths with respect to the probability of being asymptomatic, the transmission probability, and also the total number of care homes. We have plotted the relevant mappings, using output data from a large number of additional simulations. These plots illustrate that our model is not highly sensitive to independent variations in these parameters. Therefore, since the primary objective of this article is to compare interventions against the baseline scenario, with the interventions being implemented on top of the baseline model in a very natural way, we do not expect our final conclusions to be highly sensitive to small variations in these parameters. A more sophisticated analysis is currently out of reach, due to the long runtime of the model.

2. `How was the vaccine efficacy implemented in the model not very clear? Did the authors refer to the vaccine efficacy in this paper as efficacy against confirmed Covid-19 case or efficacy against transmission? What state variables of agents were affected if they are vaccinated? (Did the vaccinated agents have their state of infection changed from Susceptible to Recovered/Immune at the probability p1 and p2?) How did vaccination affect the transmission if susceptible vaccinated individuals come into contact with infected individuals and their disease progression if infected?'

Response: The subsection on vaccination, appearing in the methods section, has now been rewritten to make this all much clearer. Efficacy in our model refers to the probability of being protected against infection, after receiving a dose of the vaccine. The health of protected individuals is blocked from making the state transition Susceptible to Exposed. Protected individuals are neither able to catch the virus nor transmit it, unless they are infected at the time of vaccination in which case they are still able to transmit it.

3. `The number of simulations (ten) performed for each scenario seem quite small for a highly stochastic model like this model. I appreciate the extensive runtime for this high-resolution agent-based model. However, how did you decide that 10 simulations were sufficient to capture the true behaviours of the system? Would reducing the resolution and increasing the number of simulations produce outcomes with less uncertainty?'

Response: We have now increased the number of simulations for the baseline scenario from 10 to 100, as illustrated in Fig 17 and Fig 18, with several hundred additional simulations added to other subsections. The output distributions are concentrated around the mean and the sample variance is low, with the variance being much higher for low-resolution scaled versions of the model. This is a feature of the model that we observed early in the project, so given constraints on both time and resources, we decided on a small number of high resolution simulations, as opposed to a large number of low resolution simulations. Moreover, simulating with a high time resolution was necessary to properly implement the interventions and to capture brief encounters taking place outside the home, work or school.

4. `I appreciate that it is a very sophisticated and detailed model. However, would the purposes of the study be achieved with a simpler model such as a stochastic compartment model or an ABM with simpler structure? The authors mentioned that some of their results were similar to results from other studies which using simpler models. In this case, what would be the justification for building a more complex model?'

Response: Some of the purposes of the study could have been achieved with a simpler model, however only a detailed agent-based model is capable of supporting the full range of scenarios that we wanted to investigate. As the pandemic unfolded, the detail in the model was useful, and this shaped some of the design. With a stochastic compartmental model, or a simpler ABM, it would have been much more difficult to model all of the interventions simultaneously and realistically. The structure of our code is actually quite simple, and allows for the interventions to be modelled in a transparent and intuitive way. For example, instead of representing a lockdown using a reduction in a contact rate parameter, as is done in some models, in our model a lockdown simply directs agents home, which is more intuitive. Moreover, our model is able to capture a much higher level of heterogeneity than would be possible using a simpler compartmental model. This is an important point since these heterogeneities result in significant differences in output, as shown in Fig 19 and Table 8 of our revised manuscript.

Reviewer #2

1. `The overall text is well written but sometimes exaggerates in length, and many paragraphs could be replaced by a diagram, a table, etc, reducing the length of the text and yet improving the readability of the results.'

Response: Several paragraphs have now been reduced in length, and some omitted altogether. We have introduced a number of additional tables, as suggested, in particular Tables 7, 8, 9, 10 and 11. We agree that the text was in places rather verbose, so several paragraphs have been rewritten more concisely.

2. `There is a problem with the location and quality of the images in the review text that harmed the reading and understanding of the text. All images appeared at the end of the text, which is inopportune. I suggest fixing this issue for the next review rounds.'

Response: The images appeared at the end of the text to meet the requirements of the journal, however it does make the article difficult to read, so in the revised version we have reinserted the images directly into the text. This should also avoid any loss in image quality, which we suspect occurred during the image submission process. Moreover, we have converted our figure files using the PACE digital diagnostic tool, as suggested by the editor.

3. `In the State of Art section I think is missing a lot of recent machine learning and ABM models. This section can be improved with a deeper revision of recent research of agent-based epidemiological simulation of COVID-19.'

Response: We have now performed a much deeper review of relevant literature and have essentially rewritten the State of Art section. We now refer to a much broader range of literature, including many agent-based models of COVID-19 not referred to at all in the original version, and have inserted a paragraph specifically focussed on machine learning and epidemic modelling of COVID-19. Overall, 49 new references have been added to the manuscript.

4. `In the Methods section, the authors focused on the semantic meaning of the system components, which is perfectly understandable given the scope of the research. However, a little bit more details about the software implementation could improve the text and help the readers interested in implement or even employ the model. On the other hand, some descriptions of the model behavior (e. g. the school and classes descriptions) could be abbreviated. Or you simply share the link of the source code or you should remove from the text the promise of opening the code in the future.'

Response: Our code is now freely accessible on GitHub, together with all the documentation and input data required for other users to run the simulations. In addition, we have added some equations to the Methods section to help clarify the exposition, while abbreviating some other paragraphs, including the school and classroom descriptions.

5. `The authors let it clear in the text that the ABM has a network structure. But the Locations subsection gives the impression that the software is grid-oriented. How explicitly occur the relationships between the locations and the agents? How are they represented? Along with the text, the authors recovered the notion of a network model but in a fashion hard to understand. A good example is a car using by the agents. There will be a unique car location shared by all agents? I think that including a Network Diagram in this section, showing a small sample of the agents and their interactions will help the readers to understand better how your model works.'

Response: We agree that the talk of networks was somewhat misleading, so we have rewritten parts of the Introduction and the Locations subsection accordingly. In particular, unlike some agent-based models, our model is not based on networks. The Spatial Distribution and Location Choice subsections have also been rewritten, to help clarify the relationship between agents and locations and the role played by the grid (when initializing the random environment, locations are assigned coordinates to a resolution of 1m, first by selecting a 1km grid square and then by sampling uniformly within that square, with this being the only role played by the grid). To address the final point about network diagrams, we have now included a new figure, Fig 7, which for three randomly selected agents plots their home and personal locations on the map of the region. The line segments in Fig 7 simply illustrate the distance between an agent's home and the other locations, the agents being able to travel between these locations in any order.

6. `The ODD protocol in Appendix has an interesting high-level description of the whole system. However, I still miss some logical diagrams of the software structure and architecture (maybe a UML package diagram, or a component diagram ?)'

Response: We hope that the releasing of our code in a documented repository satisfies this requirement. We can provide a UML package diagram or a component diagram if required, possibly in exchange for relaxation of any length limits.

7. `In Model Evaluation and Validation, much of these paragraphs can be simplified or synthesized using mathematical formulas or replaced by a table with the parameters and their sources.'

Response: This section was excessively wordy in places, so has now been simplified with some paragraphs removed entirely.

8. `The results are very interesting but are spread in text and should be synthesized in a table, with the results of the proposed model and the SEIR model side by side, together with a statistical significance test of the difference between them.'

Response: We have addressed this comment with Tables 7, 8, 9, 10 and 11. Table 8 puts the results of the two models side by side, together with a statistical significance test of the difference between them, as requested.

Having made these revisions, carefully addressing each of the helpful comments raised in the review, we feel that the quality of our manuscript has been substantially improved and present the revised version for your consideration.

Yours sincerely,

James Thompson

---

## [Decision Letter · Decision Letter 1]

11 Nov 2021

PONE-D-21-09294R1Estimating the impact of interventions against COVID-19: from lockdown to vaccinationPLOS ONE

Dear Dr. Thompson,

Thank you for submitting your manuscript to PLOS ONE. After careful consideration, we feel that it has merit but does not fully meet PLOS ONE’s publication criteria as it currently stands. Therefore, we invite you to submit a revised version of the manuscript that addresses the points raised during the review process.

The reviewers recommends that you make minor revisions to your manuscripts, including further  justification of the modelling method used . Please attend to all of them and resubmit the revised one as advised in this manuscript.

We look forward to receiving your revised manuscript.

Kind regards,

Martin Chtolongo Simuunza, PhD

Academic Editor

PLOS ONE

Journal Requirements:

Reviewers' comments:

Reviewer's Responses to Questions

**Comments to the Author**

1. If the authors have adequately addressed your comments raised in a previous round of review and you feel that this manuscript is now acceptable for publication, you may indicate that here to bypass the “Comments to the Author” section, enter your conflict of interest statement in the “Confidential to Editor” section, and submit your "Accept" recommendation.

Reviewer #1: (No Response)

2. Is the manuscript technically sound, and do the data support the conclusions?

Reviewer #1: Yes

3. Has the statistical analysis been performed appropriately and rigorously? 

Reviewer #1: Yes

4. Have the authors made all data underlying the findings in their manuscript fully available?

Reviewer #1: Yes

5. Is the manuscript presented in an intelligible fashion and written in standard English?

Reviewer #1: Yes

6. Review Comments to the Author

Reviewer #1: Some further comments following the authors' responses:

1/ Regarding the vaccination efficacy, the authors only consider the all-or-nothing scenario that appears not to be the case for COVID-19 vaccines. Expermiments that consider the efficacy in terms of both infection and transmission would be more useful.

2/ I found the justification on the appropriateness of agent-based models compared with other types of stochastic compartment models still not persuasive. If the intention of the model is to compare the relative effectiveness between interventions and the authors predict that the conclusions are not highly sensitive to parameter changes, how would adding more heterogeneities into the model would result in significant differences in output. Also, if a sophisicated analysis is out of reach due to the long runtime of the model, adding more heterogeneties means adding more uncertainty into the model without being able to manage such uncertainty. This soulds like a disadvangtage of ABM.

7. PLOS authors have the option to publish the peer review history of their article (what does this mean?). If published, this will include your full peer review and any attached files.

Reviewer #1: No

---

## [Author Response · Author response to Decision Letter 1]

24 Nov 2021

1. `Regarding the vaccination efficacy, the authors only consider the all-or-nothing scenario that appears not to be the case for COVID-19 vaccines. Experiments that consider the efficacy in terms of both infection and transmission would be more useful.'

Response: Our model of vaccination, despite taking into account such factors as vaccine hesitancy, limits on daily doses, a delay between first and second doses, and the order in which individuals are vaccinated, does indeed implement vaccine efficacy in a relatively simple way. Immune responses are in reality very diverse and complex, and vary significantly between individuals, with some individuals being unable to mount an immune response at all. We therefore agree that it would be extremely useful to develop a more detailed model of immunity, and implement it in the simulations. For this reason, we are currently working on a project that aims to do just that, using a model that distinguishes immunity to infection from immunity to transmission, as suggested by the reviewer, and also immunity against symptomatic illness or severe disease. This new project also considers factors such as partial immunity, waning immunity, cross immunity for multiple variants and multiple vaccines. We feel that a deeper exploration of the topic of COVID-19 vaccination and immunity requires its own project, with many of the aforementioned factors being beyond the scope of the submitted manuscript. The all-or-nothing scenario is assumed for simplicity, with this assumption often being found in the literature. We have nonetheless revised the conclusion to our manuscript to emphasise, in response to the reviewer's comment, that the all-or-nothing approach is a limitation of the study.

2. `I found the justification on the appropriateness of agent-based models compared with other types of stochastic compartment models still not persuasive. If the intention of the model is to compare the relative effectiveness between interventions and the authors predict that the conclusions are not highly sensitive to parameter changes, how would adding more heterogeneities into the model would result in significant differences in output. Also, if a sophisticated analysis is out of reach due to the long runtime of the model, adding more heterogeneities means adding more uncertainty into the model without being able to manage such uncertainty. This sounds like a disadvantage of ABM.'

Response: While it is true that stochastic compartmental models have some advantages, agent-based models reflect the fact that epidemics do in fact result from the actions and interactions of individuals, offering a natural 'bottom-up' methodology to complement the 'top-down' methodology of the equation-based compartmental models.

Our model features heterogeneous behaviour and age, since these are clearly important factors in determining the outcome of a COVID-19 epidemic, but on the other hand does not distinguish agents by sex or by the presence of underlying medical conditions, and keeps sub-models of public transport and leisure relatively simple. Overall, our aim was to find a reasonable balance between detail and flexibility.

An agent-based model need not have more parameter uncertainty than a compartmental model, since many of the features that require parametrizing in a compartmental model, for example the flows of population between locations, are emergent in the agent-based model, consequences of the rules imposed on agents rather than the values assigned to certain variables. In an agent-based model, these rules are often intuitive, allowing for realistic representations of policy interventions. For example, in the agent-based model, lockdowns simply direct agents to stay at home (unless they are exempt), while in a compartmental model, a lockdown is represented more abstractly as a reduction in `beta', the contact rate multiplied by the transmission probability. This reduction in `beta' then requires its own estimate, a step not required in the agent-based model. Agent-based models, when implemented efficiently, can actually reduce uncertainty when compared to a heavily parametrized compartmental model.

Moreover, if it turns out that certain additional heterogeneities really do not have a big impact on the outcome, then this is in itself an interesting result, that might be difficult to obtain using a stochastic compartmental model.

We should point out that, in the case of our model, the limiting factor is not so much the complexity of the model, but rather the fact that it is written in Python, which executes relatively slowly. For this reason, we are currently working on performance enhancements that will reduce the runtime from several hours to several minutes, ultimately allowing for a more sophisticated statistical analysis of uncertainty. This is, however, still a work in progress.

We have revised our manuscript to include further discussion of the limitations of the modelling approach, and how we intend to address them in the future.

---

## [Decision Letter · Decision Letter 2]

1 Dec 2021

Estimating the impact of interventions against COVID-19: from lockdown to vaccination

PONE-D-21-09294R2

Dear Dr. Thompson,

We’re pleased to inform you that your manuscript has been judged scientifically suitable for publication and will be formally accepted for publication once it meets all outstanding technical requirements.

Kind regards,

Martin Chtolongo Simuunza, PhD

Academic Editor

PLOS ONE

Additional Editor Comments (optional):

Reviewers' comments:

Reviewer's Responses to Questions

**Comments to the Author**

1. If the authors have adequately addressed your comments raised in a previous round of review and you feel that this manuscript is now acceptable for publication, you may indicate that here to bypass the “Comments to the Author” section, enter your conflict of interest statement in the “Confidential to Editor” section, and submit your "Accept" recommendation.

Reviewer #1: All comments have been addressed

2. Is the manuscript technically sound, and do the data support the conclusions?

Reviewer #1: Yes

3. Has the statistical analysis been performed appropriately and rigorously? 

Reviewer #1: Yes

4. Have the authors made all data underlying the findings in their manuscript fully available?

Reviewer #1: Yes

5. Is the manuscript presented in an intelligible fashion and written in standard English?

Reviewer #1: Yes

6. Review Comments to the Author

Reviewer #1: I am satisfied with the revised version of the papers as the authors have addressed all of my concerns.

7. PLOS authors have the option to publish the peer review history of their article (what does this mean?). If published, this will include your full peer review and any attached files.

Reviewer #1: No

---

## [Editor Report · Acceptance letter]

9 Dec 2021

PONE-D-21-09294R2 

Estimating the impact of interventions against COVID-19: from lockdown to vaccination 

Dear Dr. Thompson:

I'm pleased to inform you that your manuscript has been deemed suitable for publication in PLOS ONE. Congratulations! Your manuscript is now with our production department. 

Kind regards, 

on behalf of

Dr. Martin Chtolongo Simuunza 

Academic Editor

PLOS ONE